# A large language model framework for sample-free population synthesis

**Michael Jones**◍*, **Richard Dawson**◍, **Jon Mills**

School of Engineering, Newcastle University, Newcastle, United Kingdom

* m.jones24@newcastle.ac.uk

## Abstract

Synthetic populations provide the demographic foundations for agent-based models in transport, public health, disaster management and other sectors, enabling credible representations of individual characteristics and behaviours. Many established synthesis methods rely on census microdata; however, such data are infrequently collected, privacy-restricted, and usually available only as small public-use samples at coarse geographic scales. This paper introduces a sample-free framework that uses a large language model (LLM) to generate complete, household-structured populations directly from aggregate demographic data. The framework is LLM agnostic and follows a multi-step process: objective definition, input preparation, LLM selection, and synthetic household generation. No model fine-tuning is required, meaning that data requirements are low and the framework is easily accessible. Population synthesis is formulated as an iterative prompting process in which an LLM generates households guided by the discrepancies between synthetic and target distributions. The model draws on prior knowledge encoded during pre-training to propose plausible attribute combinations, resulting in both statistical alignment and structural feasibility. In a global evaluation covering 109 countries, the framework achieved very close alignment on simpler marginals such as gender (SRMSE: 0.003) and household size (SRMSE: 0.026), while more structurally complex attributes such as household composition (SRMSE: 0.062) and age (SRMSE: 0.128) were also reproduced with good accuracy. These results were supported by detailed case studies in Newcastle upon Tyne (UK) and Dar es Salaam (Tanzania). The principal contribution of the framework is to enable the construction of coherent household-structured populations when detailed microdata are unavailable, expanding the applicability of agent-based modelling in data-constrained settings.

## 1 Introduction

Urban simulations are frequently founded on agent-based models (ABMs), which represent people and households as individual agents and trace how their

**Data availability statement:** All data and code supporting this study are publicly available. Processed marginals, prompts, configuration files, and example outputs are archived in the Newcastle University data repository (https://doi.org/10.25405/data.ncl.31830205). The population generation library is available at https://github.com/MJones235/LLM-Population-Generator/releases/tag/v1.0.0 and data collection and processing scripts at https://github.com/MJones235/Synthetic-Population-Experiments/releases/tag/v1.0.0.

**Funding:** MJ was funded by the EPSRC Centre for Doctoral Training (CDT) in Geospatial Systems (ref EP/S023577/1). The funder did not play any role in the study design, data collection and analysis, decision to publish, or preparation of the manuscript. URL: https://gtr.ukri.org/projects?ref=EP/S023577/1.

**Competing interests:** The authors have declared that no competing interests exist.

behaviours and interactions generate system-level outcomes. To initialise these models, researchers must first construct a population: a statistically consistent, digital representation of people, their demographic attributes, and household groupings. Synthetic populations are particularly valuable where detailed person-level records (microdata) are unavailable, incomplete, or protected for privacy reasons. They provide the demographic foundations that allow ABMs to capture heterogeneity – for instance, differences in age, mobility, or caring responsibilities – and to explore how these differences shape collective outcomes [1]. This approach has been used in many domains, including urban mobility [2–4], public health [5–9] and disaster management [10]. Across these applications, the credibility of the results depends directly on how well the synthetic population reflects the target environment.

Two primary types of data are used to construct synthetic populations: aggregate data, which provide marginal or joint distributions of demographic variables, and microdata, which contain individual-level records [11]. National censuses are the most comprehensive provider of both, but they are collected infrequently, usually on a decadal cycle, as recommended by the UN [12]. Moreover, access to person-level records is often restricted for privacy protection: only small public-use samples of 1–5% of the population are usually available, and these are restricted to coarse geographic units [13]. Where available, auxiliary surveys such as Demographic Health Surveys or Multiple Indicator Cluster Surveys can supply aggregate data, yet these sources differ in coverage, definitions and spatial resolution [14,15]. The resulting inconsistencies make it difficult to reconcile information across datasets and scales. Consequently, the central challenge is to construct household-consistent synthetic populations when the underlying data is incomplete, contradictory, or missing at fine geographic levels.

In population synthesis, the goal is to create a population whose joint distribution of demographic and household characteristics reflects the real world as closely as possible. Methods must balance three core demands: statistical fidelity (the ability to reproduce observed aggregates); feasibility (the avoidance of implausible household structures); and diversity (full coverage of possible values, including rare combinations) [16]. Early approaches focused on adjusting microdata samples to match known marginal totals, while more recent models employ probabilistic or generative techniques to infer unseen combinations of attributes.

Existing methods require a data-rich environment with extensive microdata and cross-tabulations to be effective. However, many regions around the world lack reliable, detailed demographic data, particularly in informal settlements, remote rural communities, and areas affected by poverty, conflict, or political instability [17,18]. In such contexts, it is important to generate plausible populations that make best use of the limited data available, while recognising that a perfect fit to an unknown distribution is unfeasible. This motivates the development of methods that can synthesise complete, coherent populations from partial information, rendering them usable in both data-rich and data-scarce applications.

The framework proposed in this paper employs a large language model (LLM) as a generative engine that incrementally constructs the synthetic population. Guided by

available aggregate statistics, the LLM generates individuals and households in an iterative process, continually adjusting the growing population to match target distributions. When the input data contain gaps or inconsistencies, the model draws on its prior knowledge to propose plausible combinations of demographic attributes, enabling a coherent and complete population even under limited data. The main contributions of this paper are as follows:

1. LLM-based generation: We present a sample-free generative framework that uses an LLM to iteratively create individuals and households that reflect aggregate demographic targets.

2. Empirical evaluation: We evaluate the framework through detailed case studies in Newcastle upon Tyne (UK) and Dar es Salaam (Tanzania) and test its transferability to other international contexts.

3. Open tools: We publish a documented Python codebase for both the population generation library (https://github.com/MJones235/LLM-Population-Generator/releases/tag/v1.0.0) and the data collection and processing scripts (https://github.com/MJones235/Synthetic-Population-Experiments/releases/tag/v1.0.0).

## 2 Background

Population synthesis methods can be categorised along three complementary axes: (i) computational paradigm – synthetic reconstruction (SR), combinatorial optimisation (CO), or statistical learning (SL); (ii) input data requirements – sample-based or sample-free; and (iii) output structure – individuals or households [11]. These categories provide a basis for comparing methods and for selecting an approach suited to the available data, number of demographic attributes, and the intended population scale [19].

Fig 1 organises key methodological developments along these axes. Early SR and CO methods are grouped under *sample-based fitting* to represent classical methods that reweight or draw from microdata samples. More recent *sample-free* variants extend these paradigms to contexts where only aggregate information is available. Within SL, graphical and mixture models are distinguished from the more recent emergence of deep generative models.

### 2.1 Classical approaches

Classical approaches to population synthesis work by fitting synthetic microdata to known marginal distributions. SR methods, such as iterative proportional fitting and its variants [20,21], adjust contingency tables so that they match published marginals. These methods are transparent, computationally efficient, and have been used for national-scale applications in countries such as Switzerland and Canada [22,23]. However, they are constrained by the quality and completeness of their input data: attribute combinations that do not appear in the sample cannot be generated, and results are sensitive to errors or biases in the reference microdata [20,24].

CO treats population synthesis as a search problem. It selects records from a microdata sample to build a population that best matches the target marginals [25]. This approach allows constraints to be defined more flexibly but is computationally demanding, which limits its application to smaller populations [19]. Moreover, because it depends on detailed microdata, it is most suitable for data-rich contexts.

### 2.2 Sample-free approaches

To address the challenge of generating populations where microdata are unavailable, several *sample-free* methods have been developed. These approaches rely only on aggregate statistics rather than individual-level data. Early work by Gargiulo et al. [26] introduced an iterative process that builds households member by member. A pool of individuals is first created, with ages drawn from the marginal distribution, and households are then formed using conditional rules, such as the relationships between age and household size. This method reproduces observed distributions more accurately than sample-based approaches [24] but becomes increasingly complex as more attributes are added.

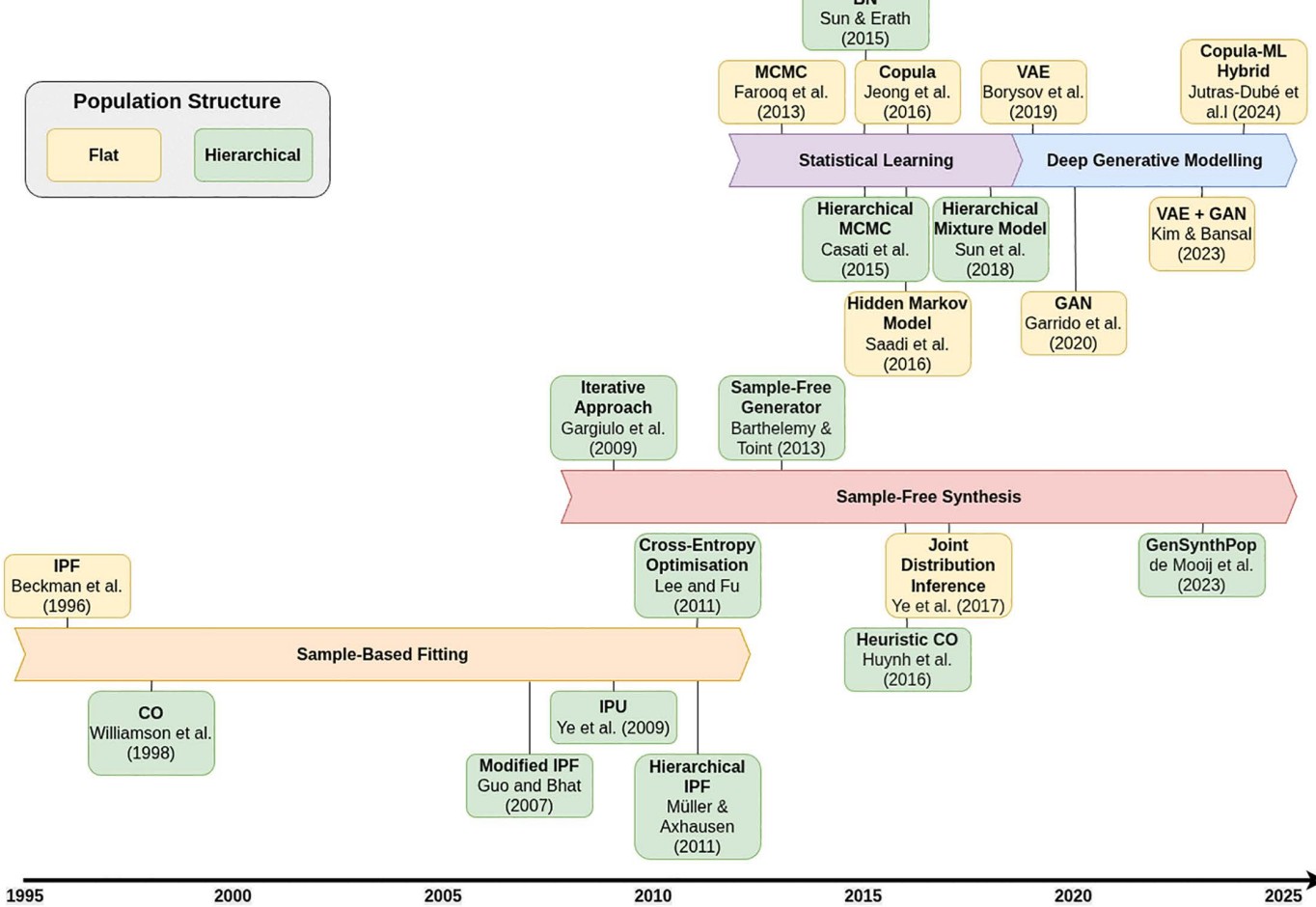

**Fig 1. Overview of key methodological advancements in synthetic population generation.** Note: IPF = Iterative Proportional Fitting; CO = Combinatorial Optimisation; IPU = Iterative Proportional Updating; MCMC = Markov Chain Monte Carlo; BN = Bayesian Network; VAE = Variational Autoencoder; GAN = Generative Adversarial Network; ML = Machine Learning.

Barthelmemy and Toint [27] extended the sample-free principle to combine data from multiple spatial resolutions. Their method draws attributes from the most detailed tables and estimates missing relationships using an entropy-based optimisation technique that balances inconsistencies between data sources. This approach improved flexibility and accuracy but required substantial data preparation and computing time. Huynh et al. [28] increased efficiency by dividing the process into two stages: first assigning core household members using fixed composition rules, then iteratively filling remaining places to ensure that all marginal totals were met. More recently, de Mooij et al. [29] proposed *GenSynthPop*, which builds populations deterministically by adding attributes sequentially to each individual. Each new attribute is conditioned on those already assigned, maintaining consistency across multiple distributions and allowing additional attributes to be easily appended. Although effective, this approach still relies on the availability of detailed cross-tabulations.

## 2.3 Statistical learning

SL methods estimate the joint distribution of population attributes from observed microdata. Once this distribution is fitted, new synthetic individuals can be generated by sampling from it. Bayesian networks express dependencies between

variables in a graphical structure, where links indicate conditional relationships such as the probability of being employed given age or education [30]. Mixture models extend this concept by assuming that the population can be grouped into different classes, within which attributes are modelled jointly [31]. This allows individual- and household-level characteristics to be fitted simultaneously. Copula-based methods build joint distributions by preserving the relationships between attributes when adjusting to new marginal distributions, offering improved dependence preservation compared to IPF [32]. Other implementations use simulation-based sampling, such as Markov Chain Monte Carlo techniques, which repeatedly draw from conditional probability tables until the synthetic population reproduces the overall joint patterns observed in the training data [33,34]. Such methods can reproduce realistic combinations of attributes that are absent from the microdata but depend on detailed, internally consistent inputs and are computationally intensive.

More recent deep generative models build on the same principle but replace explicit probability structures with neural networks that learn high-dimensional dependencies directly from microdata. Architectures such as variational autoencoders and generative adversarial networks have been adapted for population synthesis, learning flexible joint distributions from microdata and then sampling entirely new individuals and households [16,35,36]. These models can generate plausible combinations of attributes absent from the sample but may also create unrealistic or infeasible records, such as contradictory household compositions, if not properly constrained [16].

### 2.4 LLMs

LLMs are increasingly used as tools for synthetic data generation [37–39]. Trained on vast text corpora, they learn statistical dependencies between words and can generate coherent outputs in response to written prompts [40]. LLMs extend the statistical learning paradigm: like deep generative models, they approximate the joint distribution of variables, but they do so implicitly through patterns learned from language rather than from domain-specific datasets.

The GReaT framework [41] demonstrates that LLMs can act as tabular data generators. In this framework, each row in a dataset is stringified and used to fine-tune a LLM, which can subsequently be used to generate new synthetic samples [41]. Similar approaches have been applied to population data, demonstrating improved feasibility and comparable diversity to deep generative methods [42]. These studies confirm that LLMs can learn dependencies between demographic attributes from microdata, but the approach remains fundamentally sample-based. It requires access to detailed individual records, repeated fine-tuning for each region or dataset, and produces only flat individual records without household structure.

A second line of research explores prompt-based generation, where pre-trained LLMs are directed through textual instructions rather than additional training. This approach uses the model's existing linguistic and contextual knowledge to produce data that reflect specified relationships or constraints. Ling et al. [43] demonstrated this with the MALLM-GAN framework, which iteratively refines prompts in an adversarial setup to improve the fit between synthetic and real tabular data.

This paper extends existing research on LLM-based population synthesis by introducing a sample-free approach that operates solely on aggregate distributions, offering the potential for application to a wider range of geographies and sectors. Existing LLM frameworks, such as GReaT and MALLM-GAN, are sample-based, relying on microdata for fine-tuning or conditioning, while traditional sample-free methods require detailed cross-tabulations. In contrast, this approach leverages the relationships already encoded with a LLM's pre-training data, allowing such dependencies to be captured implicitly. The framework is intentionally simple, requiring no fine-tuning or complex adversarial architectures, and demonstrates that a LLM can generate hierarchical populations directly from aggregate targets.

## 3 Methodology

### 3.1 Overview

The proposed framework uses a LLM to iteratively construct individual and household records that align with specified demographic targets. At each step, the model is guided by a prompt containing both the current and target distributions for each variable, enabling it to adjust outputs dynamically to reduce statistical error. The process is entirely sample-free and

 

relies only on aggregate data. The framework comprises a workflow consisting of three main steps, as summarised in . The first step establishes the synthesis objectives and prepares the required inputs, including the target distributions. The second step formulates the prompt and selects a suitable LLM by evaluating candidate models on a small test case. The third step carries out the population synthesis itself through repeated querying of the model.

Together, these steps produce a complete synthetic population that is both structurally valid and statistically consistent with the input targets. The following subsections describe each step in detail.

### 3.2 Problem formulation and objectives

Synthetic population generation seeks to construct a population of artificial households and individuals that reproduces a set of observed statistical descriptors. Let a synthetic population, $P$, be defined as

$$P = \{h_1, h_2, \ldots, h_N\}, \; h_i = \{p_{i1}, p_{i2}, \ldots, p_{in_i}\}$$

where each household $h_i$ is a collection of $n_i$ individuals. Each individual $p_{ij}$ is represented by a vector of attributes

$$p_{ij} = \left(x_{ij}^{(1)}, \ldots, x_{ij}^{(M)}\right) \in X$$

where X denotes the complete set of valid attribute combinations. Some household-level descriptors such as size and type may be derived from the attribute members, rather than being assigned directly:

$$\text{size}\,(h_i) = n_i, \; \text{type}\,(h_i) = \phi\left(\left\{\left(x_{ij}^{(relationship)}, x_{ij}^{(age)}\right)\right\}_{j=1}^{n_i}\right)$$

where $\phi(\cdot)$ maps sets of intrahousehold relationship and ages to categorical household types.

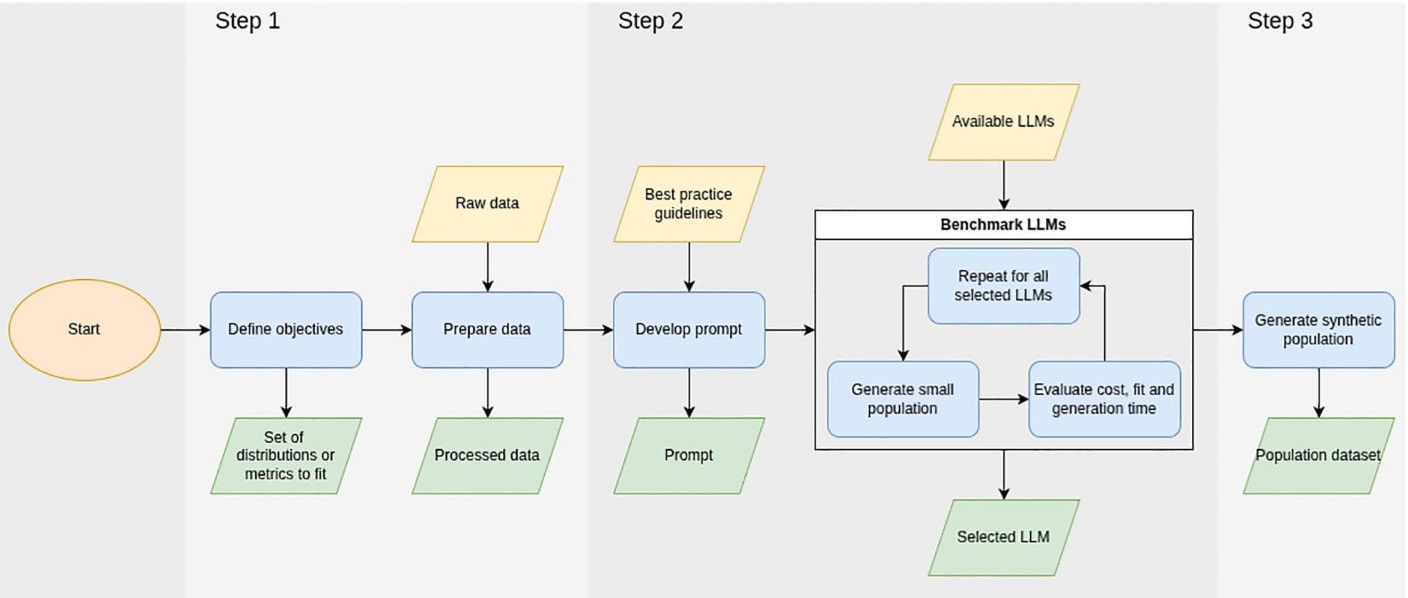

**Fig 2. Schematic of the LLM-based generation framework.**

For a given study region, let $D = \{D_k\}_{k=1}^{K}$ denote the available target descriptors. Each $D_k$ may be a marginal distribution, a low-order cross-tabulation, or a summary statistic such as a mean value. The corresponding empirical statistics calculated from synthetic population $P$ are written $G_k(P)$. The objective of the synthesis task is to find a population $P \in \mathcal{F}(S, \mathcal{R})$ that minimises the total discrepancy across all descriptors:

$$\min_{P \in \mathcal{F}(S,\mathcal{R})} \sum_{k=1}^{K} \delta_k \left( G_k(P), D_k \right)$$

(1)

where $\delta_k$ denotes a discrepancy measure appropriate to descriptor $D_k$, such as root mean square error. The feasible set $\mathcal{F}(S, \mathcal{R})$ enforces schema constraints $S$ and structural rules $\mathcal{R}$, which are discussed further in Section 3.3. The discrepancy functions $\delta_k$ are used both to guide the generation process (see Section 3.4) and to evaluate the quality of the outputs (see Section 3.5).

This formulation frames population synthesis as a constrained optimisation problem, where the model must generate records that both respect structural feasibility and reduce statistical error relative to the specified marginals.

## 3.3 Input preparation

To operationalise the synthesis objective defined in Section 3.2, three inputs must be supplied: (i) a set of aggregate descriptors $D = \{D_k\}$, (ii) a schema $S$ specifying the structure of the synthetic population, and (iii) a set of rules, R, to enforce more complex constraints. These inputs provide the statistical targets to be matched and the feasibility rules to be enforced during population generation.

Aggregate descriptors are typically drawn from census data or other household surveys. They may include marginal distributions (e.g., age or household size), joint distributions (e.g., age-by-gender), or scalar statistics such as means. The framework is designed to accommodate a wide range of descriptor types and imposes no requirement for complete or consistent coverage. When reliable data are not available for a particular variable, that variable can be excluded from the targets and treated as unconstrained. This design allows the framework to operate flexibly across both data-rich and data-scarce environments.

All descriptors must be transformed into a two-column CSV format with one column for the category name and one for the corresponding percentage share of the population. This structure is required to standardise the prompt construction and validation routines.

The schema $S$ is defined in JavaScript Object Notation (JSON) format and specifies the attributes to be generated along with their valid values and data types. For instance, it may specify that age should be an integer between 0 and 120 years. Structural rules $\mathcal{R}$ are used to capture more complex requirements, such as the minimum age for household heads or allowed relationship combinations. Both $S$ and $\mathcal{R}$ are applied during validation to ensure that generated records form logically coherent households.

## 3.4 Generation algorithm

Population synthesis is performed as an iterative process in which the selected LLM generates candidate households in batches, guided by feedback as to how well the current population aligns with the target descriptors $D = \{D_k\}$. At each iteration, a prompt is constructed around four core elements:

1. Task instructions specifying the synthesis goal and format requirements;

2. Schema excerpt defining valid attribute values and data types;

3. Target distributions $D_k$ provided as tabulated reference statistics;

4. Current synthetic distributions $\hat{D}_k$, computed from the partial population generated so far.

An example prompt is reproduced in Appendix A (S1 File).

The prompt is submitted to the selected LLM, which returns the candidate household. The household is parsed and validated against the schema $S$ and structural rules $\mathcal{R}$. Invalid records are rejected and returned to the model along with a diagnostic message prompting correction. Valid households are added to the synthetic population $P$, and the process continues until the specified number of households is reached.

The approach implements a feedback-driven loop in which the LLM iteratively adjusts its sampling behaviour based on discrepancy information embedded in the prompt. The full procedure is summarised in Algorithm 1.

**Algorithm 1. Iterative, feedback-driven synthesis of a population.**

```
Input: target descriptors D, schema S, feasibility rules R,
population size N, batch size B, large language model fθ
Initialise P←∅
while |P|<N do
    Compute empirical descriptors Ď from current population P
    Construct prompt including: task instructions, schema excerpt,
    reference targets D, and current empirical distributions Ď
    Query fθ with the prompt to generate B candidate households
    for each household h in candidates do
        if h satisfies schema S and rules R then
            add h to P
        else
            return error message to fθ and request corrected household
            if corrected household valid then add to P
        end if
    end for
end while
Output: synthetic population P
```

The iterative feedback design was adopted following a preliminary experiment in which households were generated without information about the population accumulated so far. Despite the prompt containing target distributions, fit was poor: attribute selection probabilities did not align with targets. Full details are provided in Appendix B (S1 File). These results confirmed that target distributions alone are insufficient to guide generation, motivating the feedback-driven architecture described above.

### 3.5 Evaluation framework

The synthetic population is evaluated along two core dimensions: distributional fit and feasibility. These criteria reflect the framework's ability to produce a population that is both statistically aligned with input targets and internally consistent with defined structural constraints.

**3.5.1 Distributional fit.** Distributional fit measures how closely the generated data replicate the target statistical descriptors $D = \{D_k\}$. For each descriptor $D_k$, the corresponding empirical distribution $G_k(P)$ is computed from the synthetic population $P$, and the discrepancy $\delta_k\left(G_k(P), D_k\right)$ is calculated. Three metrics are used to quantify this error. The primary measure is the Standardised Root Mean Square Error (SRMSE), one of the most widely used measures of statistical alignment in population synthesis research [16,22,30]. For a one-dimensional categorical descriptor with observed proportions $q_i$ and target proportions $p_i$.

$$SRMSE(P, Q) = \sqrt{n \sum_{i=1}^{n} \left(p_i - q_i\right)^2}$$

where $n$ is the number of categories. Lower SRMSE values indicate stronger alignment, with 0 corresponding to a perfect match. SRMSE is computed for all fitted marginals and serves as the primary measure of goodness-of-fit.

As a secondary measure, Jensen-Shannon Divergence (JSD) is reported. This metric provides a symmetric, information-theoretic measure of divergence between two discrete probability distributions. It is defined as:

$$JSD(P, Q) = \frac{1}{2} \sum_{i=1}^{n} p_i \log_2 \frac{p_i}{m_i} + \frac{1}{2} \sum_{i=1}^{n} q_i \log_2 \frac{q_i}{m_i}, \quad m_i = \frac{1}{2} (p_i + q_i)$$

JSD is bounded between 0 and 1, where 0 indicates identical distributions. Unlike SRMSE, which is unbounded and sensitive to the number of categories, JSD provides a normalised fit measure that is comparable across marginals.

For ordered variables, such as age and household size, the Wasserstein distance $W_1$ is also reported. Unlike SRMSE and JSD, which treat categories as exchangeable, $W_1$ accounts for the ordinal structure of the distribution. It measures the minimum cost of transforming one distribution into the other, where cost is proportional to the distance mass moved.

$$W_1(P, \ Q) = \sum_{i=1}^{n} |F_P(i) - F_Q(i)|$$

where $F_P$ and $F_Q$ are the empirical cumulative distribution functions of the synthetic and target distributions respectively. All categories are treated as equally spaced ordinal units; open-ended terminal bins such as '80+' are therefore assigned the same width as closed bins, which may slightly underestimate the true transport cost in the upper tail.

**3.5.2 Feasibility.** Feasibility refers to whether each household satisfies the structural constraints defined by the schema $S$ and the ruleset $\mathcal{R}$. Let

$$I\left(h_i; \mathcal{S}, \mathcal{R}\right) = \begin{cases} 1, & \text{if } h_i \text{ satisfies schema } \mathcal{S} \text{ and rules } \mathcal{R} \\ 0, & \text{otherwise} \end{cases}$$

The overall feasibility rate is then

$$F = \frac{1}{N} \sum_{i=1}^{N} I(h_i; S, \ \mathcal{R})$$

representing the proportion of households that are valid.

## 3.6 LLM selection

The framework is model-agnostic and compatible with any LLM capable of structured generation. However, different models vary in their ability to reason over statistical constraints, produce valid outputs, and maintain consistent formatting. Selecting an appropriate model is therefore a necessary preparatory step before population synthesis.

Model selection is performed through a short benchmarking procedure. Each candidate model is tested using a fixed prompt template with the task of generating a small population sample of 800 households. The outputs are evaluated on three dimensions: (i) distributional fit to the target descriptors (measured using JSD), (ii) structural feasibility of the generated records (proportion of valid households), and (iii) computational efficiency (runtime and token cost). The model offering the best overall balance across these criteria is selected for use in the full synthesis process.

The candidate models evaluated are pre-trained foundation models, and no domain-specific fine-tuning is performed within this framework. This is a deliberate choice to ensure that the data requirements remain low and the framework remains accessible.

## 4 Case studies

Three experiments were designed to evaluate the proposed framework under contrasting data environments. The objective was to assess its performance across diverse demographic contexts and to test robustness when the available descriptors range from complete to highly fragmented. Each case study provides a distinct data landscape, allowing examination of how the framework behaves under different levels of information quality and consistency.

The first experiment tested global applicability and checked for regional bias. Age and gender data were drawn from the World Population Prospects dataset [44], using the most recent 2023 estimate for all countries. Household data were taken from the Database on Household Size and Composition [45], which collates and harmonises data from heterogeneous sources (e.g., Multiple Indicator Cluster Surveys, Demographic Health Surveys, Integrated Public Use Microdata Series). To limit quality issues, any data from before 2010 or with >0.1% unknown values were excluded. After filtering, the sample covered 108 of the 193 UN member states. Additional household data was sourced from the 2021/2 UK Census, bringing the total to 109 [46]. For each country, a small population of 500 households was generated using identical prompts.

The second experiment focused on Newcastle upon Tyne (UK), a data-rich urban environment in which all descriptors were available from the 2021 Census [47]. Because the inputs were complete, internally consistent, and drawn from a single source, Newcastle served as a benchmark case to assess the framework's accuracy under ideal conditions. The resulting population represented a baseline for evaluating the effects of data sparsity in the third and final experiment.

The third experiment considered Dar es Salaam (Tanzania), where data availability was limited and fragmented across multiple sources. As summarised in Table 1, key variables originated from datasets collected at different times and geographic aggregation levels. This case therefore tested the model's ability to synthesise coherent populations when inputs were inconsistent or partially incompatible.

Table 2 summarises the demographic attributes and category groupings used in all experiments. Rather than harmonising the data between case studies, it was decided to retain the attribute values used in the original data sources. This demonstrated the ability of the method to utilise additional information, where available.

Household composition was derived from explicit role relationships in conjunction with household size and individual age, as detailed in Table 3.

For Newcastle and Dar es Salaam, larger populations of approximately 100,000 individuals were generated to enable detailed evaluation of internal structure and feasibility. Generation proceeded in batches of ten households until the target size was reached, using identical synthesis parameters. The global samples were generated with the same procedure but limited to 500 households per country for efficiencies in both time and financial cost.

## 5 Results

### 5.1 LLM benchmarking and selection

Nine LLMs were benchmarked to select the most appropriate model for population generation. Each model was tasked with generating an 800-household synthetic population using an identical prompt template, given in Appendix A (S1 File).

**Table 1. Input data for Dar es Salaam.**

| Variable | Dataset | Data Type | Geographic Level |
|---|---|---|---|
| Age by gender | Tanzania Population Estimates and Projections, 2015–2030 [48] | Estimated (for 2025 from 2022 Census) | Dar es Salaam |
| Average household size | Tanzania Demographic and Health Survey and Malaria Indicator Survey 2022 Final Report [49] | Measured | Dar es Salaam |
| Household size | Tanzania Demographic and Health Survey and Malaria Indicator Survey 2022 Final Report [49] | Measured | Urban Tanzania Mainland |
| Household composition | Household Size and Composition [45] | Estimated (from 2015 DHS) | Tanzania |

**Table 2. Summary of variables from the survey data.**

| Attribute | Values | | | Notes |
|---|---|---|---|---|
| | Global | Newcastle | Dar es Salaam | |
| Gender | Male; female | Male; female | Male; female | As reported in all sources |
| Age group | 0-9; 10-19; …; 70-79; 80+ | 0-9; 10-19; …; 70-79; 80+ | 0-9; 10-19; …; 70-79; 80+ | 5-year age bands aggregated to 10-year groups |
| Household size | 1; 2-3; 4-5; 6+ | 1; 2; …; 7; 8+ | 1; 2; …; 8; 9+ | As reported in all sources |
| Household composition | One-person; couple only; couple with children; lone parent; extended family; non-relatives; unknown | One-person aged 66+; one person aged <66; couple; couple with dependent children; couple with non-dependent children; lone parent; other | One-person; couple only; couple with children; lone parent; extended family; non-relatives; unknown | Overlapping categories collapsed (e.g., lone mother/father/parent collapsed to lone parent) |

**Table 3. Mapping of household relationship structures to UK and global classifications.**

| Household structure | UK classification | Global classification |
|---|---|---|
| Head only (head age < 66) | One person aged <66 | One-person |
| Head only (head age ≥ 66) | One-person aged 66+ | One-person |
| Head and partner | Couple | Couple |
| Head and 1 + child | Lone parent | Lone parent |
| Head, partner and 1 + child; any child age < 18 | Couple with dependent children | Couple with children |
| Head, partner and 1 + child; all children age ≥ 18 | Couple with non-dependent children | Couple with children |
| Head and a combination of relatives not covered above (e.g., grandparent, aunt, cousin) | Other | Extended family |
| Head and 1 + housemate | Other | Non-relatives |
| Any other combination | Other | Non-relatives |

All experiments were run on a standard workstation (AMD Ryzen 9 3900X, 64GB RAM, NVIDIA GeForce GTX 980), with temperature and top-p fixed at 1.0 across all experiments.

Table 4 summarises distributional fit (mean JSD), structural feasibility (error modes, failures and success rates), and computational efficiency (generation time) for each model. The success rate reports the percentage of households generated within the allowed retry limit; each household was attempted up to three times, and error counts were aggregated across all attempts. JSON errors denote malformed or non-parseable outputs; schema errors indicate valid JSON that violates the predefined structure (e.g., missing fields); and API errors capture infrastructure-level failures (e.g., timeouts).

Three additional domain-specific validation rules were applied: each household must contain exactly one head; no minors may live alone; and age-relationships consistency must be maintained (e.g., parents must be older than children). No violations of these domain-specific rules were observed for any model, indicating that the LLM respected structural constraints without requiring rule-based correction. The validation step therefore acted primarily as a safeguard against formatting errors rather than structural infeasibility.

The small, locally hosted models (*Llama3.2:1-3b* and *Phi3-mini*) performed poorly across all dimensions: distributional fit was low, failure rates were high (primarily due to invalid JSON outputs), and inference times were impractically long for large-scale synthesis on consumer-grade hardware. The reasoning-orientated models (*Deepseek-r1-0528, o3-mini, Grok-3-mini*) achieved the lowest JSD values, indicating superior distributional alignment, but at substantially higher inference times.

**Table 4. Distributional fit, structural feasibility and generation time for 9 LLMs on an 800-household test case.**

| Model | Version | Access method | Time (min) | Mean JSD | JSON errors | Schema errors | API errors | Failed households | Success rate (%) |
|---|---|---|---|---|---|---|---|---|---|
| GPT-4o | 2024-11-20 | OpenAI on Azure | 19 | 0.049 | 5 | 3 | 0 | 0 | 100.0 |
| GPT-4o-mini | 2024-07-18 | OpenAI on Azure | 22 | 0.224 | 0 | 1 | 0 | 0 | 100.0 |
| Deepseek-r1-0528 | 1 | Azure AI Foundry | 772 | 0.008 | 677 | 0 | 0 | 132 | 83.5 |
| o3-mini | 2025-01-31 | OpenAI on Azure | 129 | 0.008 | 0 | 0 | 0 | 0 | 100.0 |
| Grok-3 | 1 | Azure AI Foundry | 28 | 0.037 | 0 | 0 | 0 | 0 | 100.0 |
| Grok-3-mini | 1 | Azure AI Foundry | 503 | 0.009 | 1 | 10 | 4 | 0 | 100.0 |
| Llama3.2:1b | 3.2:1b | ollama | 703 | 0.362 | 1127 | 1162 | 28 | 785 | 1.9 |
| Llama3.2:3b | 3.2:3b | ollama | 264 | 0.313 | 465 | 1237 | 1 | 556 | 30.5 |
| Phi3-mini | 3:mini | ollama | 1087 | 0.331 | 1728 | 16 | 592 | 772 | 3.5 |

Table 5 reports token usage and cost for each model. Differences in mean input token counts across models reflect two parameters: re-prompting overhead for models with higher failure rates, where corrective prompts included the original response and error message; and differences in tokenisation strategy across model families, which affect how the same text is segmented into tokens. Cost per household for the paid models ranged from low (*GPT-4o-mini* at 0.01p) to prohibitive for large populations (*Deepseek-r1-0528* at 2.44p), with the high cost of reasoning models driven primarily by their large output token counts.

Rankings were broadly consistent across JSD, SRMSE and Wasserstein distance (see Table C1, Appendix C, S1 File), though some metric-dependent variation was observed. For age, *GPT-4o-mini* achieved a lower Wasserstein distance than *Llama 3.2 3B* (0.48 vs 1.03), suggesting better ordinal fit, yet its SRMSE was higher (0.88 vs 0.59), indicating greater category-level deviation. These divergences illustrate the different aspects of error captured by each metric but do not affect the model selection decision.

Overall, *GPT-4o* was judged to provide the best balance between distributional fit, speed and reliability. It consistently produced valid outputs and outperformed its lighter variant (*GPT-4o-mini*) by a wide margin in distributional accuracy, as illustrated in Figure C1, Appendix C (S1 File). *Grok-3* also performed well, with slightly better distributional fit than *GPT-4o*, but slightly higher generation time and cost. *GPT-4o* was used in all further experiments.

**Table 5. Token usage and cost for 9 LLMs on an 800-household test case. Costs are correct as of September 2025.**

| Model | Mean input tokens/ prompt | Mean output tokens/ prompt | Total input tokens | Total output tokens | Cost per 1M input tokens (£) | Cost per 1M output tokens (£) | Cost input (£) | Cost output (£) | Total cost (£) | Cost per household (p) |
|---|---|---|---|---|---|---|---|---|---|---|
| GPT-4o | 965 | 68 | 779,570 | 54,803 | 1.85 | 7.39 | 1.44 | 0.40 | 1.84 | 0.23 |
| GPT-4o-mini | 1009 | 62 | 808,482 | 49,638 | 0.11 | 0.44 | 0.09 | 0.02 | 0.11 | 0.01 |
| Deepseek-r1-0528 | 928 | 3404 | 1,247,613 | 4,578,444 | 1.00 | 3.99 | 1.25 | 18.27 | 19.52 | 2.44 |
| o3-mini | 940 | 1545 | 752,050 | 1,236,310 | 0.81 | 3.25 | 0.61 | 4.02 | 4.63 | 0.58 |
| Grok-3 | 939 | 68 | 751,200 | 54,791 | 2.22 | 11.08 | 1.67 | 0.61 | 2.27 | 0.28 |
| Grok-3-mini | 912 | 64 | 743,153 | 52,140 | 0.19 | 0.94 | 0.14 | 0.05 | 0.19 | 0.02 |
| Llama3.2:1b | 1038 | 387 | 2,420,031 | 902,527 | 0.00 | 0.00 | 0.00 | 0.00 | 0.00 | 0.00 |
| Llama3.2:3b | 1061 | 102 | 2,065,827 | 199,298 | 0.00 | 0.00 | 0.00 | 0.00 | 0.00 | 0.00 |
| Phi3-mini | 751 | 149 | 1,774,403 | 352,956 | 0.00 | 0.00 | 0.00 | 0.00 | 0.00 | 0.00 |

## 5.2 Global evaluation

The performance of the proposed synthesis framework was evaluated across a globally distributed sample of countries. The aim was to assess its robustness when applied to diverse demographic contexts.

**5.2.1 Summary of distributional fit.** The performance of the framework was summarised across all 109 countries in the global sample. Distributional fit was quantified using SRMSE, calculated for each of four core demographic descriptors: age, gender, household size, and household composition. Table 6 presents descriptive statistics for SRMSE across all countries, both by individual descriptor and for the combined mean.

Performance varied substantially across descriptors. Fit was strongest for gender and household size, with low median SRMSE and limited spread. Higher errors were observed for household composition and age, reflecting the greater structural complexity of these distributions. Household composition, for instance, was inferred from the size of the household, along with the age and role of the constituent members. Meanwhile, age distributions had to be carefully balanced to ensure realistic age differences between parents, children and other family members.

For age, Wasserstein distance showed moderate agreement with SRMSE across 109 countries (Pearson $r = 0.649$, Spearman $\rho = 0.718$, $p < 0.001$), as shown in Figure D1, Appendix D (S1 File). The two metrics diverged for a subset of countries, with the direction of divergence seemingly related to the shape of the population pyramid. Countries with expansive pyramids such as Nigeria, Madagascar and Sudan, tended to have high SRMSE relative to their Wasserstein distance, as the synthetic distributions overestimated the 30s age group while underestimating the 20s – errors that were penalised more heavily by SRMSE. The opposite pattern was observed for countries with constrictive pyramids, such as Italy, Portugal and Canada, where the Wasserstein distance was relatively high despite moderate SRMSE.

**5.2.2 Spatial variation in fit.** Fig 3 maps the mean SRMSE for each of the 109 countries in the evaluation set. High-performing countries were geographically diverse, with examples across South Asia, Latin America and parts of Europe. In contrast, many of the poorest results were concentrated in Africa and the Middle East. Europe was underrepresented in the evaluation due to the nature of the input dataset, which drew primarily on surveys conducted in low- and middle-income countries. In high-income regions, only census data were available, and these were often excluded because they were outdated or reported high levels of missing information.

Fig 4 shows synthetic and target age-gender pyramids for 9 countries, selected for their geographic and demographic diversity. Population pyramids for all 109 countries in the evaluation set are provided in Figure D2-D6, Appendix D (S1 File). The best-performing countries, such as India and Indonesia, tend to have near-stationary pyramids: broadly rectangular at the bottom with smooth tapering at older ages. Mid-performing countries displayed more variation. Some, like Ethiopia, Ghana and Kenya, have expansive pyramids with large youth cohorts. Others, such as Italy and Portugal, are more constrictive, reflecting aging populations. Many African countries fell in the mid-performing group with higher median SRMSE than other continents, though as noted above, this gap narrowed considerably when assessed by Wasserstein distance.

**Table 6. Summary statistics of SRMSE for distributional fit across 109 countries.**

| Distribution | Mean | Median | Std | IQR | Min | Max |
|---|---|---|---|---|---|---|
| Age | 0.157 | 0.128 | 0.098 | 0.127 | 0.028 | 0.491 |
| Gender | 0.004 | 0.003 | 0.005 | 0.004 | 0.000 | 0.044 |
| Household Size | 0.030 | 0.026 | 0.023 | 0.012 | 0.003 | 0.173 |
| Household Composition | 0.096 | 0.062 | 0.077 | 0.093 | 0.013 | 0.384 |
| Combined Mean | 0.072 | 0.065 | 0.034 | 0.042 | 0.021 | 0.183 |

 

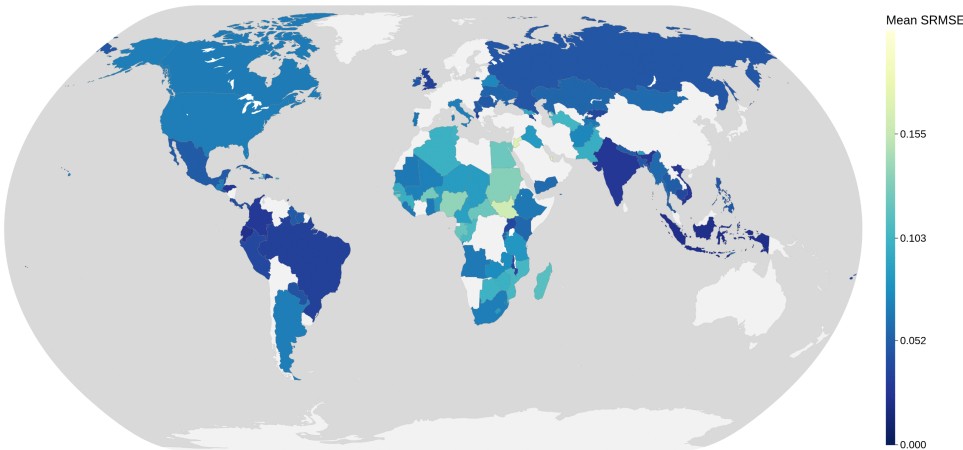

**Fig 3. Mean SRMSE across 109 countries.**

The poorest performance was seen in a small number of outliers, namely Qatar, the Maldives and Jordan. Qatar and the Maldives have highly imbalanced population pyramids with extreme male surpluses, driven by large migrant worker populations [50,51]. In Qatar, the synthetic population had too many under-20s, suggesting that the model either allocated too many children per family, or that it failed to capture adult children living with their parents. Jordan has a smaller gender imbalance but hosts a substantial refugee population, which may contribute to non-standard household and age structures [52]. In all three cases, the demographic patterns are atypical and the LLM struggled to reproduce them accurately.

To assess the consistency of the framework across runs, multiple population pyramids were generated for the best (Indonesia), median (Ghana) and worst (Qatar) performing countries. Fig 5 illustrates the variation in outputs across 20 independent runs. Mean values and ranges are shown for each age-gender category. While some variability was observed between runs, the framework consistently reproduced populations of similar shape and structure.

### 5.3 Newcastle upon Tyne, UK

Newcastle upon Tyne was selected as a benchmark to evaluate the framework under data-rich conditions. All input descriptors were drawn from the 2021 UK Census, providing a complete, internally consistent dataset. This setting represents a best-case scenario where the model's performance was not constrained by data quality or coverage. The generated population consisted of 42,500 households (97,238 individuals), approximately one third of the city's total population.

Fig 6 compares the marginal distributions for four core demographic attributes: age, gender, household size, and household composition. Fit was strong across all variables, with SRMSE ranging from 0.001 for gender to 0.158 for household composition. The largest deviation came from an overrepresentation of "couple" households and underrepresentation of "other" households. Since the total number of two-person households was correct, this would imply that the model had a slight resistance to generating households containing two unrelated housemates.

Beyond marginals, the model also captured multivariate relationships, even though these were not explicitly controlled during generation. Fig 7 plots the synthetic and target frequencies of attribute combinations across the six bivariate tabulations. It also plots a multivariate tabulation that includes all four of the core input variables. Agreement was strong across all cases with $R^2$ values ranging from 0.573 to 0.994. Highest alignment was observed in low-dimensional relationships (e.g., household size by gender), while more complex cross-tabulations, particularly those involving age and household composition, showed greater variability.

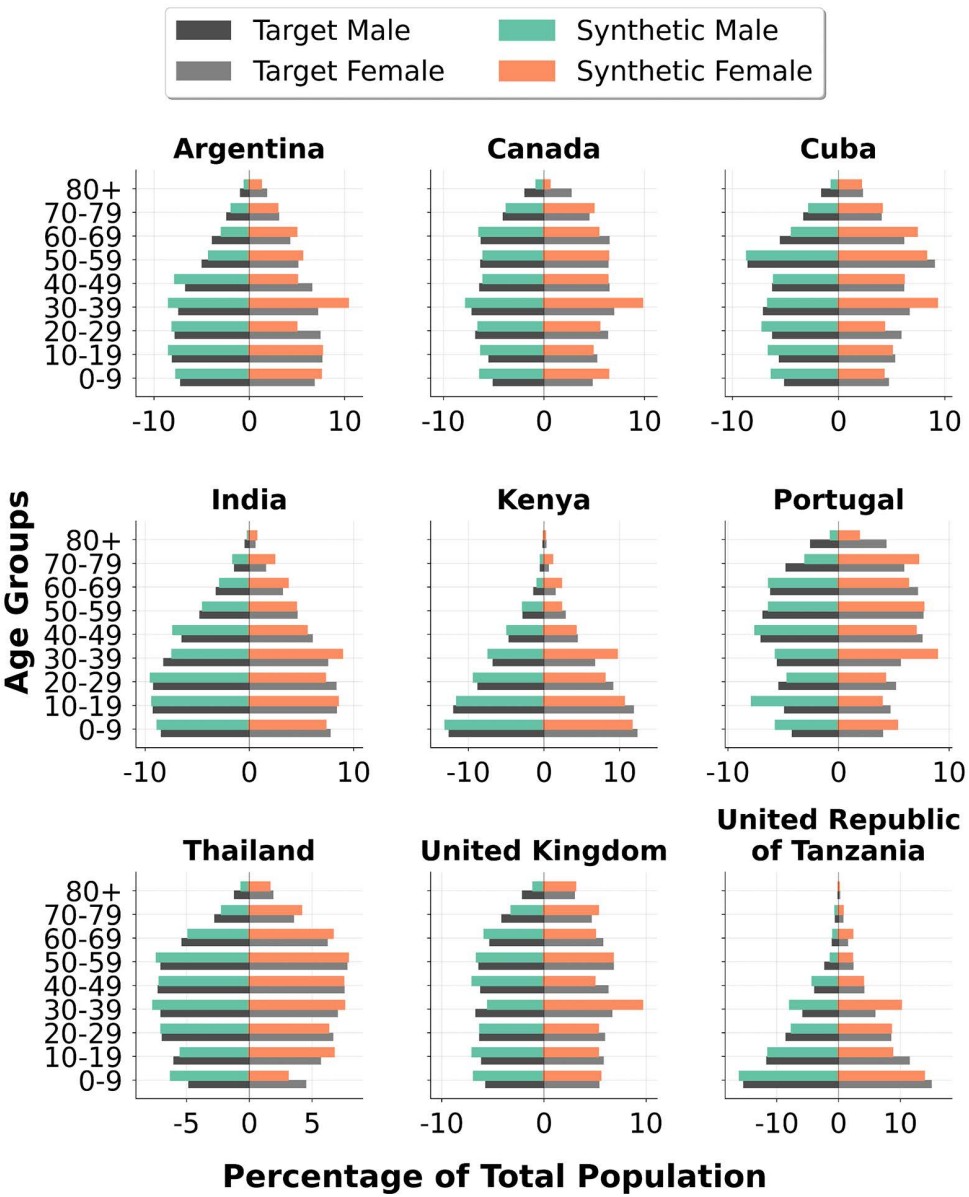

**Fig 4. Synthetic and target age-gender pyramids for 9 selected countries.**

Fig 8 evaluates age-based intra-household relationships that were not provided as inputs to the framework, providing held-out validation of the model's ability to recover emergent demographic structure. Panel (a) compares the age gap between opposite-gender partners. While the census distribution is approximately normal and centred at a one-year male-older difference, the synthetic data displayed a bimodal pattern, with peaks at ±2 years. The individual age gaps produced by the model were plausible, but the overall distribution diverged from the ground truth profile. The absence of same-age couples was a notable quirk. Panels (b) and (c) plot the age of fathers and mothers, respectively, at the birth of their child. In both cases, the synthetic values fell within plausible bounds and broadly reproduced the shape of the census data. However, there was a tendency to over generate parents in their late 20s and under generate those in their late 30s. As

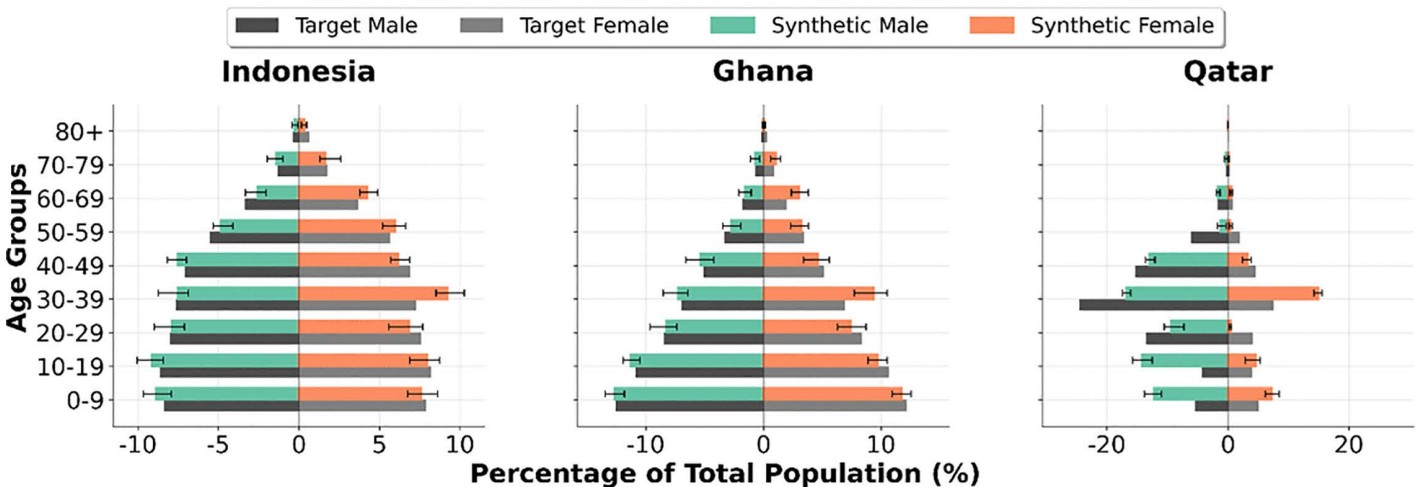

**Fig 5. Population pyramids for Indonesia, Ghana and Qatar, showing the mean and range of categorical percentages across 20 independent trials.**

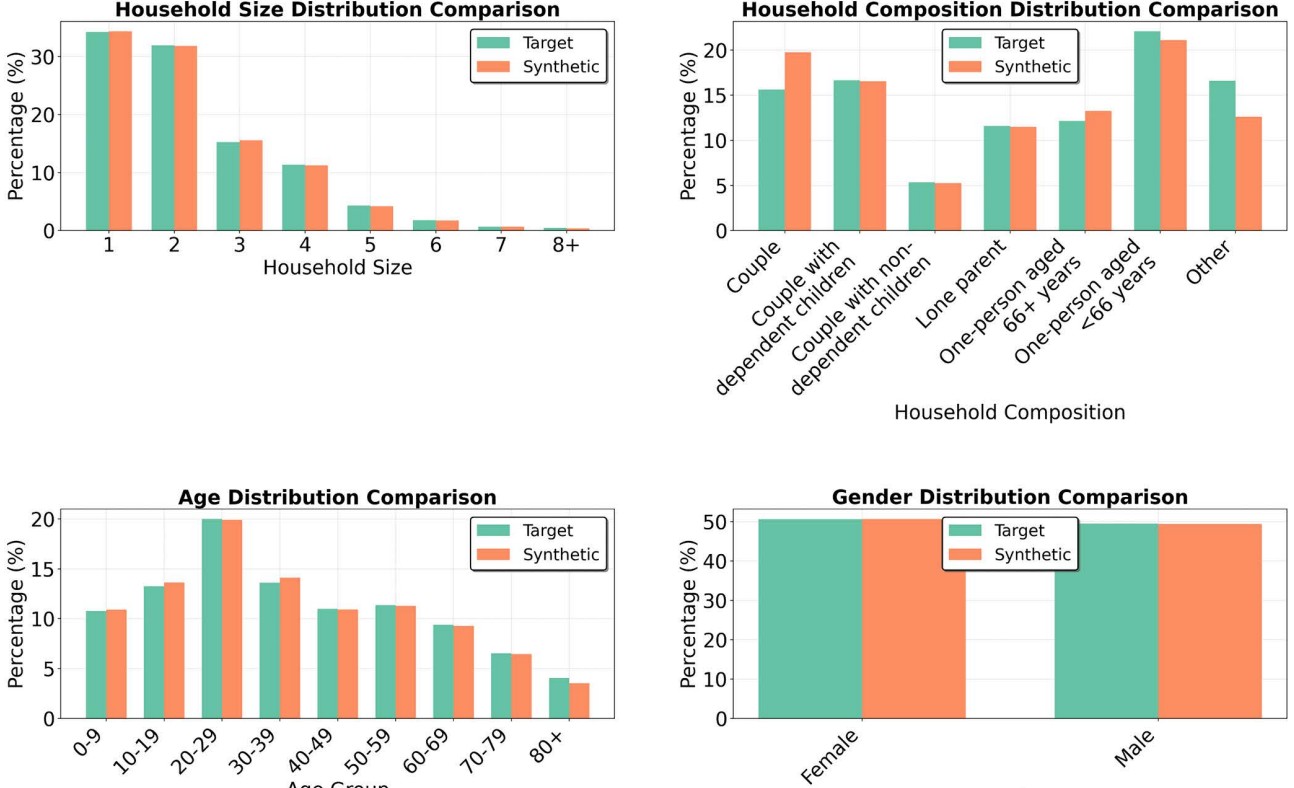

**Fig 6. Marginal distributions for Newcastle upon Tyne (note different Y axis scales).**

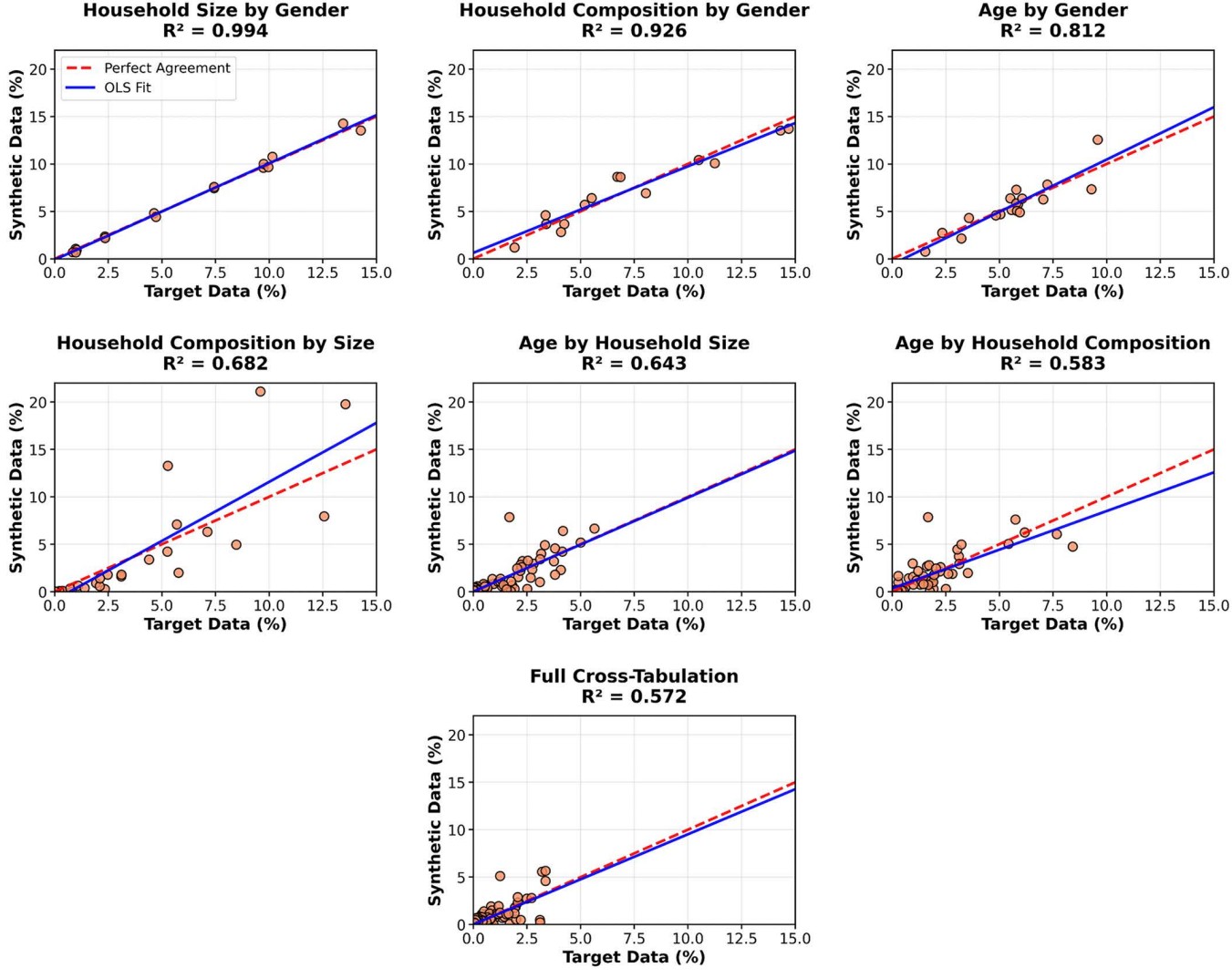

**Fig 7. Joint distributions for Newcastle upon Tyne.**

with the partner age gap, this reflects the absence of specific feedback during generation and could likely be refined by adapting the algorithm if needed for a particular application.

Fig 9 tracks SRMSE over the course of population generation. For most descriptors, error declined rapidly as the population grew, with convergence reached by around 2,000 individuals (approximately 800 households). Age distribution continued to show gradual improvement as 100,000 individuals was reached. Household composition showed higher error than the other distributions, with only marginal improvement beyond 2,000 individuals. This may imply that the LLM does not understand how to improve this distribution, perhaps not realising that two-person unrelated households are a valid combination.

To test consistency across runs, twenty independent populations of 800 households were generated. This population size was selected since the majority of SRMSE convergence has been achieved by this point. Fig 10 compares target and synthetic marginal distributions across these runs, while Table 7 summarises the corresponding SRMSE statistics. Variability was lowest for gender (mean 0.003, std 0.002) and household size (mean 0.032, std 0.009), higher for age (mean

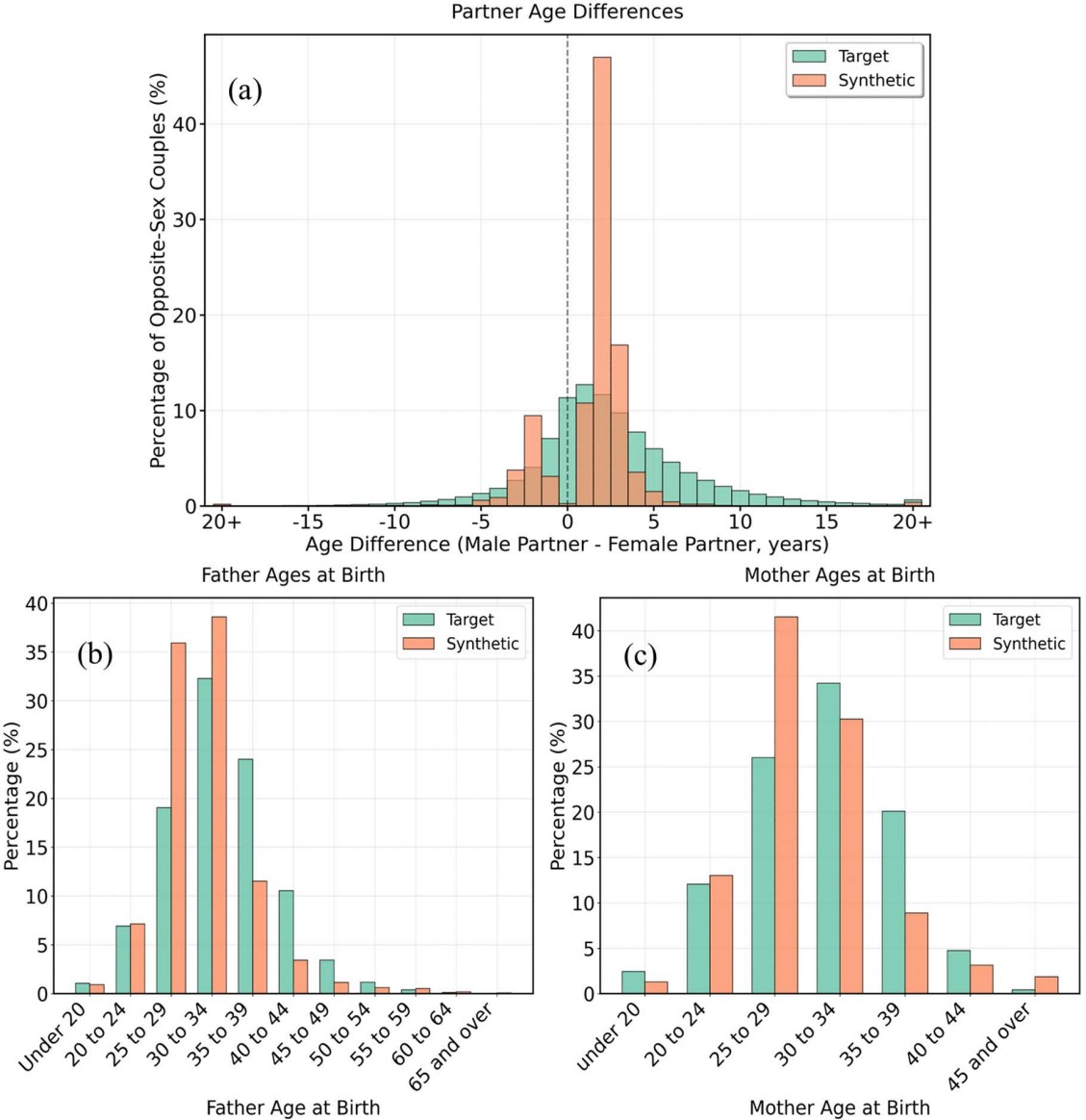

**Fig 8. Intra-household age differences.**

0.110, std 0.020), and greatest for household composition (mean 0.231, std 0.044). Age displayed a consistent tendency to overrepresent individuals aged 30–39 and underrepresent those aged 80＋. Household composition also showed systematic bias, with "couple" households overrepresented and "other" households underrepresented across all runs.

To quantify uncertainty in the $R^2$ values reported in Fig 7, a bootstrap analysis was conducted across the 20 repeated runs. As shown in Table 8, confidence intervals were broadly stable across runs for most cross-tabulations. Confidence intervals were tightest for household composition by size ($R^2$ = 0.666, 95% CI = [0.646, 0.695]) and widest for household composition by gender ($R^2$ = 0.778, 95% CI = [0.656, 0.894]). These results indicate that the multivariate structure of the synthetic population is recovered consistently, though with greater uncertainty for some attribute combinations.

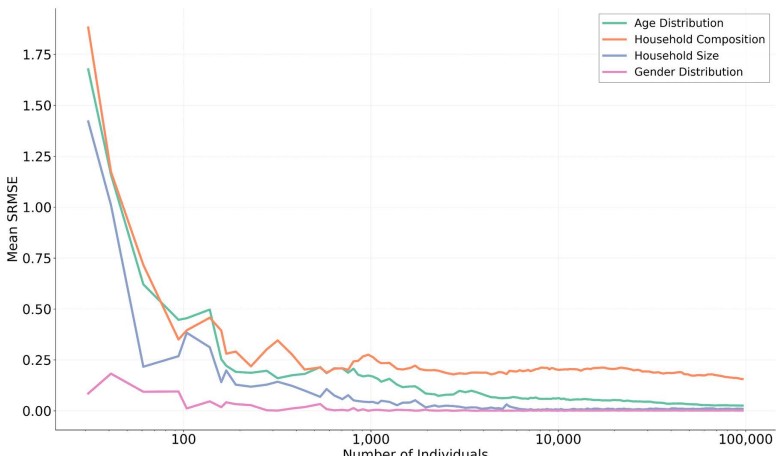

**Fig 9. SRMSE convergence as individuals are incrementally added to the synthetic population.**

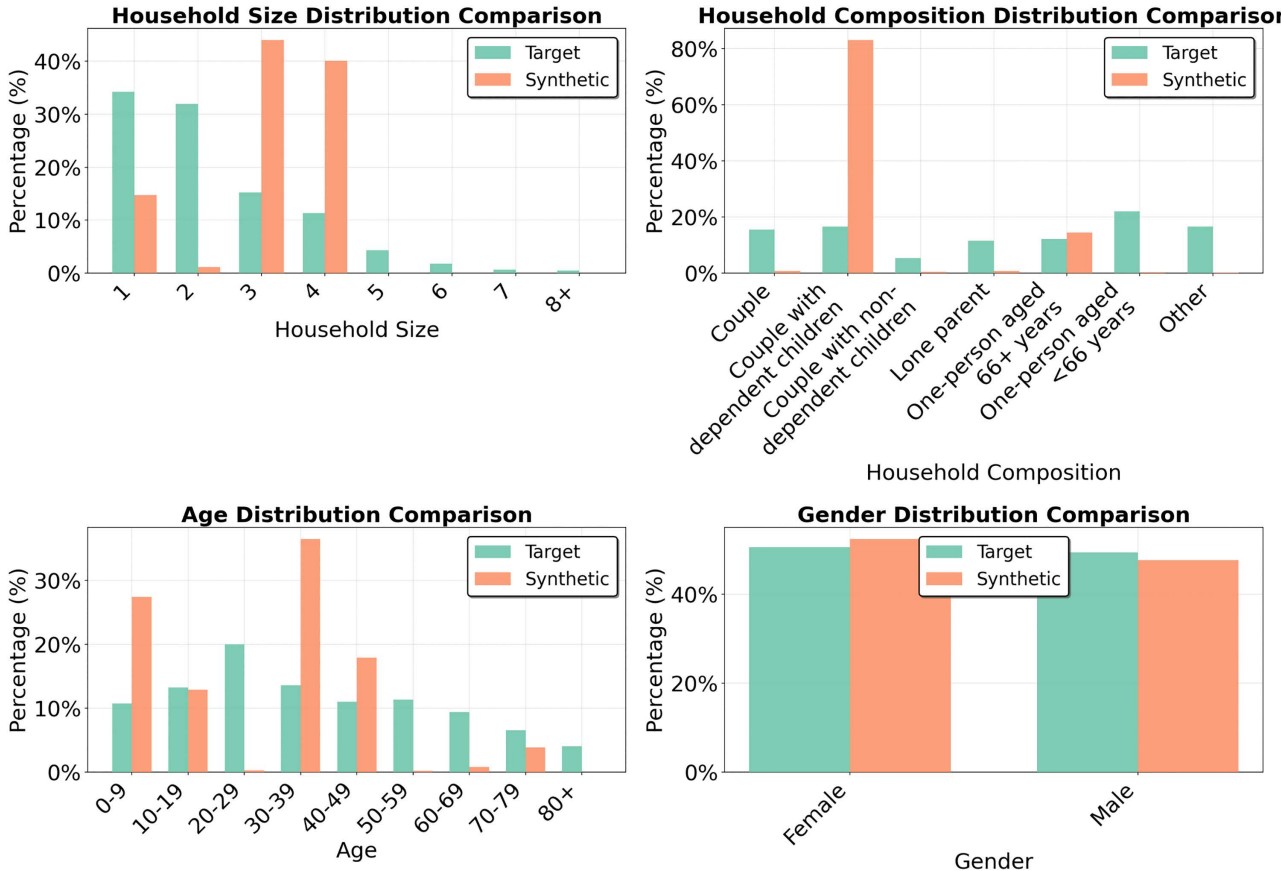

**Fig 10. Comparison of target and synthetic marginal distributions across 20 independent runs (note different Y axis scales).**

**Table 7. Summary statistics of SRMSE for distributional fit across 20 independent runs.**

| Distribution | Mean | Median | Std | IQR | Min | Max |
|---|---|---|---|---|---|---|
| Age | 0.110 | 0.107 | 0.020 | 0.034 | 0.074 | 0.145 |
| Gender | 0.003 | 0.002 | 0.002 | 0.002 | 0.000 | 0.010 |
| Household Size | 0.032 | 0.030 | 0.009 | 0.009 | 0.021 | 0.058 |
| Household Composition | 0.231 | 0.236 | 0.044 | 0.065 | 0.149 | 0.310 |

**Table 8. Bootstrap confidence intervals for $R^2$ values across cross-tabulations, based on 20 independent runs of 800 households.**

| Cross-tabulation | Categories | Mean $R^2$ | 95% CI |
|---|---|---|---|
| Household size x gender | 16 | 0.948 | [0.889, 0.989] |
| Household composition x gender | 14 | 0.778 | [0.656, 0.894] |
| Age x gender | 18 | 0.891 | [0.840, 0.932] |
| Household composition x size | 28 | 0.666 | [0.646, 0.695] |
| Age x household size | 72 | 0.633 | [0.557, 0.714] |
| Age x household composition | 53 | 0.553 | [0.465, 0.661] |
| Age x gender x household size x composition | 256 | 0.576 | [0.503, 0.644] |

## 5.4 Dar es Salaam, Tanzania

Dar es Salaam provided a contrasting case to Newcastle, offering assessment of the framework under imperfect data conditions. Here, the marginal distributions were assembled from multiple sources that varied in spatial resolution and collection period. They included at least one instance of directly conflicting values, making a perfect fit to all targets impossible. The aim, therefore, was not exact replication, but to assess whether the framework produced a coherent population when faced with inconsistent or incomplete information. To achieve this, a population of 96,775 individuals was generated across 25,000 households.

A discrepancy existed between the household size and composition marginals in their treatment of one-person households. The size distribution reported 14.5%, while the composition distribution gave 9.6%. The generated synthetic population contained 12.8%, indicating that the model implicitly resolved this conflict by converging on an intermediate value. This suggests that, in the absence of explicit prioritisation instructions, the framework treats competing marginals as equally weighted, partially satisfying each constraint rather than privileging one over the other. As shown in Fig 11, the framework reproduced individual-level attributes well, with SRMSE values of 0.082 for age and 0.009 for gender. Household-level attributes were fitted less precisely but remained within an acceptable range given the limitations of the input data. SRMSE was 0.192 for household size and 0.146 for household composition. Notably, the household composition fit was slightly better than in Newcastle, which may reflect differences in category definitions: in addition to having fewer composition categories, the Tanzania data distinguishes non-relatives from extended family, whereas the UK data groups these under a broader "other" category. In contrast, household size was less well captured, with 9 + person households representing just 0.05% of the synthetic population compared to the 3.6% target.

## 5.5 Comparison with iterative proportional updating

To contextualise the framework's performance relative to established methods, a direct comparison with Iterative Proportional Updating (IPU) was conducted for Lower Layer Super Output Area (LSOA) E01008291 in Newcastle upon Tyne, which contains 883 households. This area was selected because its size is consistent with the convergence point identified in section 5.3. A comparison was not possible for Dar es Salaam, as IPU requires microdata that are not available, making it inapplicable in the data-scarce settings that motivate the LLM approach.

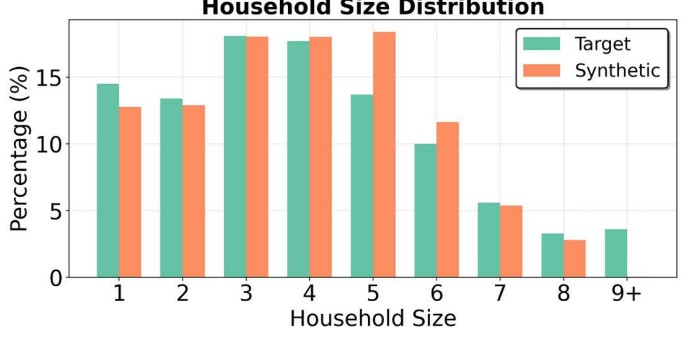
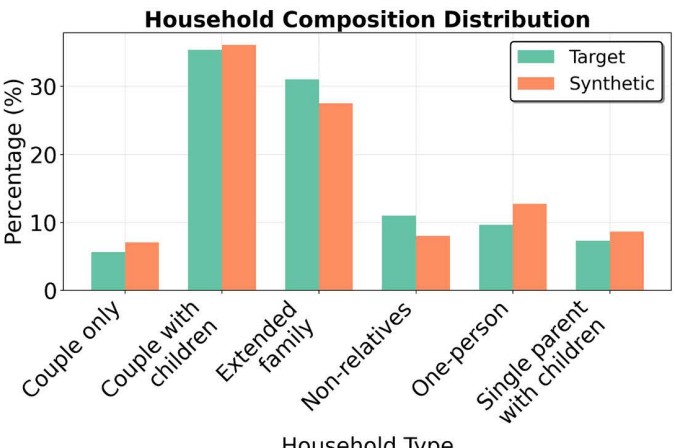
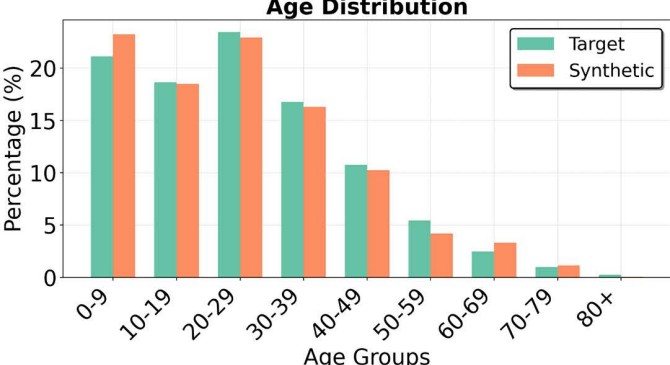
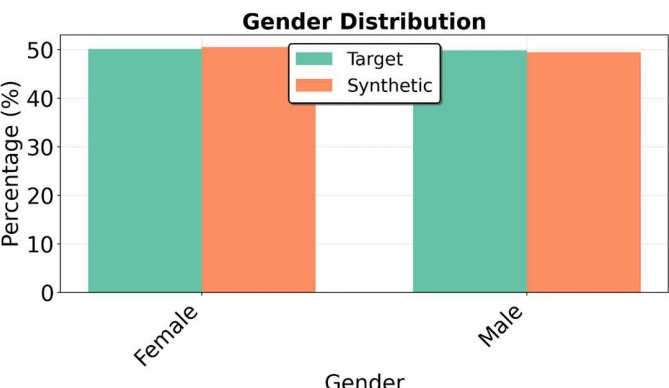

**Fig 11. Marginal distributions in Dar es Salaam for two prompt configurations (note different Y axis scales).**

**5.5.1 Methods.** IPU is a sample-based synthetic reconstruction method that extends Iterative Proportional Fitting (IPF) to jointly satisfy both individual- and household-level constraints by iteratively reweighting a microdata sample [53]. It was implemented here using the mlfit R package [54] with 1,000 iterations.

The microdata source was the 2021 UK Census 10% Safeguarded sample [55], filtered to the North East region, which was the finest geographic granularity available. LSOA-level marginal distributions were obtained from the 2021 Census and used as fitting targets for both methods. To ensure compatibility, the variables were matched to those available in the microdata: resident age in six categories (≤15; 16–24; 25–34; 35–49; 50–64; 65+), gender (male; female), household size (1–8+), and household composition in eight categories describing the number of adults and children (e.g., 1 adult; 2 adults; 2 adults and 1–2 children).

To assess the stability of each method, 20 independent populations were generated for each approach. For IPU, a different random seed was used each time. Distributional fit was evaluated using SRMSE and Wasserstein distance.

**5.5.2 Results.** Fig 12 compares the marginal distributions generated by each method across 20 independent runs, with mean values and 95% confidence intervals shown. The LLM framework achieved better fit for age (SRMSE: 0.013 vs 0.058), gender (SRMSE: 0.003 vs 0.011), and household size (SRMSE: 0.064 vs 0.074). IPU achieved better fit for household composition (SRMSE: 0.081 vs 0.206). Mean SRMSE across all four variables favoured IPU, driven by the

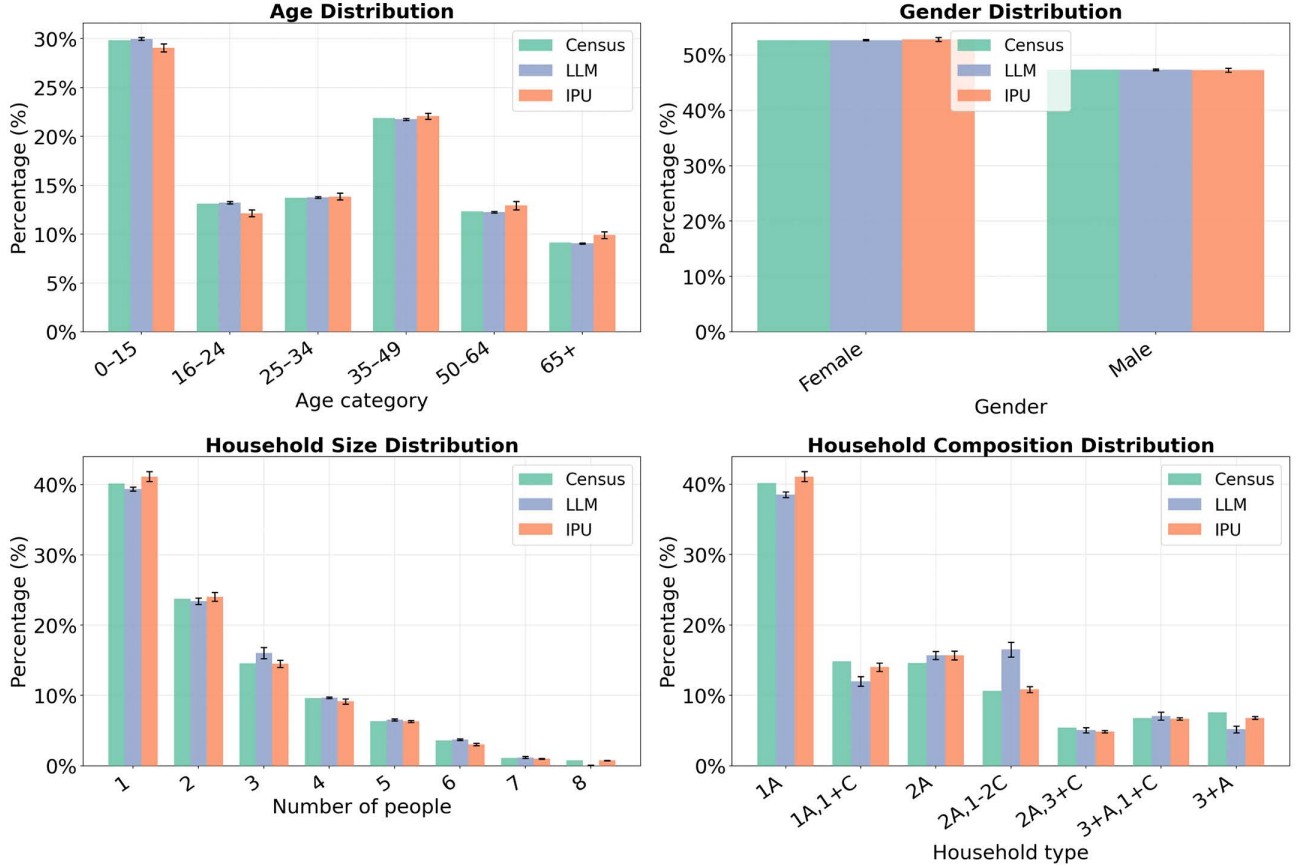

**Fig 12. Marginal distributions for LSOA E01008291 generated by the LLM framework and IPU across 20 independent runs, compared against 2021 Census targets.** Bars show mean values; error bars show 95% confidence intervals. A = adults, C = children, so a household composition of "1A,1+C" indicates one adult and one or more children.

large difference in household composition fit. When assessing using Wasserstein distance for the two ordered variables, similar relative performance was observed. Variability across runs was higher for the LLM framework, reflecting the stochastic nature of LLM generation.

Beyond distributional fit, the two methods differ in their practical characteristics. IPU was substantially faster and cheaper, completing in under 1.5 seconds at negligible cost compared to 15 minutes and £1.91 per run for the LLM framework. However, IPU is constrained to the variables and categories recorded in the microdata, meaning age is available only in six broad bands. Further, IPU requires extensive microdata for the case study location. In contrast, the LLM framework can generate either categorical or integer ages, and generates comparable performance with limited input data.

## 6 Discussion

### 6.1 Overview of findings

This study demonstrated that LLMs can generate synthetic populations that are both structurally valid and closely aligned with aggregate demographic characteristics of settlements and countries. The method is entirely sample-free, requiring only marginal distributions, and does not rely on microdata, cross-tabulations, or retraining. This sets it apart from

classical fitting methods, which rely on detailed reference samples, and from statistical learning methods, which require task-specific training and are less readily transferable across contexts.

Across experiments, the framework produced household-structured populations that preserved realistic demographic relationships. In Newcastle, where the input data were complete and consistent, distributional fit was strong for all core variables and joint patterns were accurately inferred. In Dar es Salaam, where inputs were fragmented and partially incompatible, the framework still generated a coherent and plausible population. In such settings, the goal is not perfect replication but sensible balancing of conflicting constraints. The behaviour observed shows that the framework can use available information to interpolate reasonable household patterns and maintain internal coherence.

International evaluation further confirmed that the framework generalises well across diverse demographic contexts, producing plausible populations under varying conditions. Its outputs were stable across repeated runs, and it performed reliably in both data-rich and data-scarce settings. Together, these findings establish LLM-based population synthesis as a flexible and transferable approach that expands the range of environments in which credible synthetic populations can be generated.

## 6.2 Performance across countries and settlements

Evaluation across 109 countries demonstrated strong generalisation across diverse demographic regimes. The framework reliably reproduced stationary, expansive and constrictive population pyramid shapes, with highest accuracy seen in countries such as India and Indonesia. Performance was also strong across much of Latin America and parts of Europe, confirming that the framework is not region-specific.

Performance varied across continents, with Africa showing the highest median SRMSE. However, this finding was sensitive to the choice of metric: when age distributions were evaluated using Wasserstein distance, the relative performance of African countries improved significantly, as the category-level errors were penalised less heavily.

The poorest fits were observed in a small number of outlier countries with highly atypical population structures. In Qatar and the Maldives, for example, international labour migration has produced strongly skewed age-gender profiles that deviate from standard demographic patterns. These cases suggest that outputs may be less reliable in settings where demographic structures diverge sharply from those commonly encountered in model training.

The framework's ability to reproduce household composition varied across the three experiments. Composition fit was better in both Dar es Salaam and the UK global evaluation than in the standalone Newcastle case study. The former two used the same category scheme, which distinguished between non-relative and extended family member households, while the UK data included a broad "other" category, which lacked a formal definition and may have been more difficult for the model to interpret and reproduce reliably.

The factors driving performance variation remain difficult to isolate and are likely multiple. Where input marginals are contradictory, as in Dar es Salaam, a perfect fit to all targets is logically impossible regardless of model quality. Demographic representation in the model's training data remains a plausible contributor, particularly for regions and demographic regimes that are less well represented in the large-scale text corpora. Univariate regressions against four country-level predictors (Appendix E, S1 File) did not reveal strong associations with mean SRMSE, confirming that performance is not straightforwardly predictable from observable covariates alone; disentangling the relative contributions of factors such as population composition and data recency will require more targeted experimentation in future work. Where biases persist, they may have practical consequences for downstream applications in transport, health and disaster management, where demographic structure directly shapes individual behaviour and vulnerability.

## 6.3 LLM choice

The framework is designed to be model agnostic, but in practice performance is highly dependent on the reasoning ability of the underlying language model. In this study, *GPT-4o* was selected for its favourable balance between distributional

fit, cost and inference time. Smaller models tested during development produced poor alignment with target distributions, while larger 'reasoning' models achieved better fit but were significantly slower and more expensive.

Comprehensive benchmarking was not undertaken, given the large and rapidly evolving set of available models. Frequent releases, changing APIs, and variable pricing render systematic comparison quickly out-of-date. The results presented here illustrate typical trade-offs, rather than defining an optimal configuration. As part of the framework, users are advised to benchmark available models for their specific use case, weighing accuracy, runtime and cost.

## 6.4 Sensitivity

Prompt design has a substantial influence on the quality of outputs produced by LLMs, a sensitivity that has been widely documented across language-modelling tasks [56]. Even minor adjustments in phrasing or punctuation can alter model reasoning paths, yielding measurable differences in quantitative performance. To examine the implications of this behaviour for population synthesis, a systematic prompt-sensitivity analysis was conducted.

A meta-prompt (reproduced in Appendix F, S1 File) was used to generate fifteen linguistic variants of the baseline instruction used in the previous studies. Each prompt variant was evaluated using *GPT-4o*, *GPT-4o-mini*, and *Grok-3* – selected due to their fast generation times (see Table 4) – to produce synthetic populations of 500 households per configuration. The resulting populations were compared to the target distributions using SRMSE across the four primary variables (age, gender, household size, and household composition). Fig 13 summarises the variation in fit scores across prompts and models.

Missing bars indicate that fewer than 450 of the 500 households met the feasibility criteria defined in Section 3.5.2, and the generation was therefore classified as a failure. In most cases, this was due to malformed JSON output, although some failures were caused by the selection of attributes that violated the schema constraints.

Model performance varied significantly across prompt variants. *Grok-3* showed stable behaviour, with relatively little variation in SRMSE, and was the only model to generate a valid population for all variants. *GPT-4o* exhibited greater variation and 8 out of 15 prompts failed to produce a valid population. *GPT-4o-mini* demonstrated the highest overall error and absolute variability, though only failed to generate a population on one occasion.

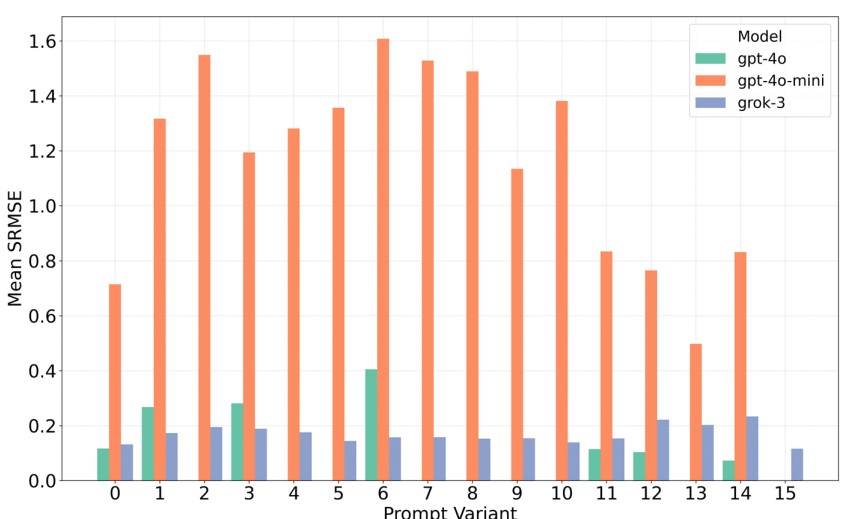

**Fig 13. SRMSE fit variability across 15 prompt variants for three different LLMs.** Prompt 0 is the baseline prompt. Missing bars indicate failed generations, defined as runs in which fewer than 450 of the 500 target households met the feasibility criteria defined in Section 3.5.2.

The relative ranking of models changed depending on the prompt. *GPT-4o* achieved the lowest SRMSE overall (0.093) under prompt variant 14 and outperformed *Grok-3* in three variants (11, 12, and 14). Conversely, *Grok-3* outperformed *GPT-4o* in variants 1, 3, and 6, with differences of up to 0.218 SRMSE. This confirms that prompt phrasing has a material impact on model output and that individual LLMs respond differently to alternative instructions.

These findings suggest that prompt selection should not be treated as fixed. We recommend testing multiple prompt formulations for each model and selecting the best-performing variant. This process should be repeated when changing models, as prompt effectiveness does not generalise across LLMs. For models such as *GPT-4o*, particular attention should be given to the structure and clarity of output formatting instructions to avoid parsing failures.

## 6.5 Batch size

The framework generates households sequentially, submitting one prompt per household by default. While this approach simplifies prompt construction and validation, it is computationally inefficient, since each call incurs fixed overhead from tokenising the instruction and schema. To assess whether generating multiple households per prompt could improve efficiency without sacrificing output quality, a systematic batch size experiment was conducted using *GPT-4o* and the Newcastle case study. Batch sizes from 1 to 35 were tested, with 10 repeated runs per configuration. Fig 14 summarises the results across six dimensions: generation time, API call volume, reliability, cost, output diversity and distributional fit.

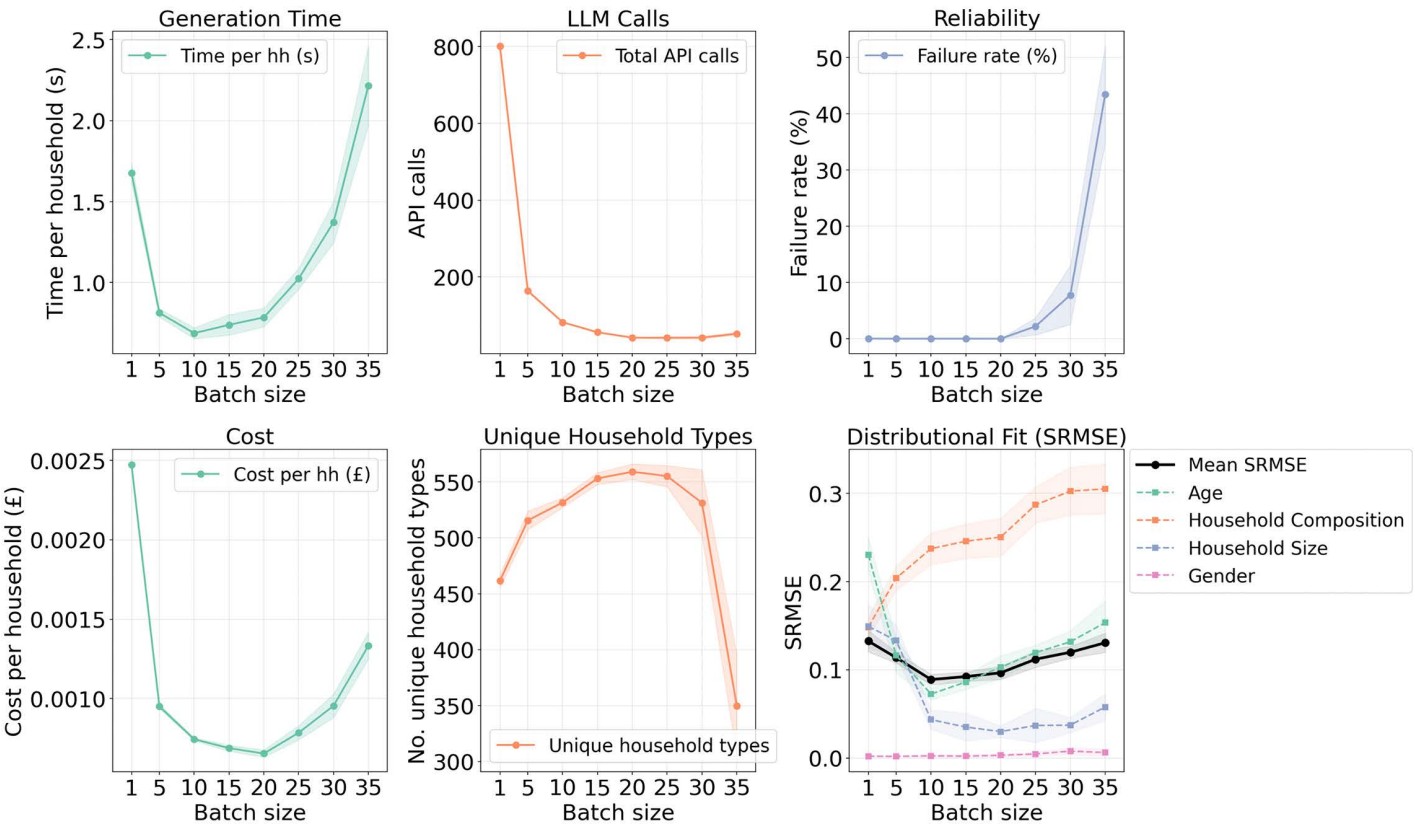

**Fig 14. Effect of batch size on six dimensions of framework performance, based on 10 repeated runs per configuration using GPT-4o and the Newcastle case study.** Lines show mean values; shaded bands show 95% confidence intervals.

---

Generation time per household followed a U-shaped profile, reaching a minimum at a batch size of around 10 before rising steeply. This pattern reflects two competing effects: at small batch sizes, the large number of LLM calls dominates; at large batch sizes, the outputs take longer to generate and failure rates increase substantially, requiring additional corrective prompts. Reliability degraded markedly beyond a batch size of 20, with failure rates exceeding 40% at a batch size of 35. These failures were primarily due to schema validation errors resulting from the incorrect number of households being generated. This resulted in an increase in both input and output tokens and, consequently, cost.

Distributional fit improved for both age and gender as batch size increased from 1 to 10, while household composition degraded with increasing batch size. Output diversity, measured as the number of unique household types, increased with batch size up to around 15–25, beyond which it declined sharply, coinciding with the onset of elevated failure rates.

Taken together, these results indicate that a batch size of around 10 offers the best balance of efficiency, reliability, diversity and distributional fit. Compared to the single-household default, it reduces generation time by 59%, cost by 70% and API call volume by 90%, while improving mean SRMSE from 0.133 to 0.089.

## 6.6 Limitations

Although the proposed framework performs well across a range of data conditions, several limitations remain. The most immediate concern relates to scalability. Because generation is handled through repeated LLM queries, both time and cost scale linearly with the number of households. This makes the approach well suited to city-scale populations typical of agent-based models, but expensive for national-scale synthesis under current configurations. For example, as of November 2025, producing a synthetic UK population of 69 million people would cost approximately £70,000 using *GPT-4o* and a batch size of 1. However, as demonstrated in Section 6.5, generating multiple households per prompt can substantially reduce both generation time and cost.

A further limitation concerns the nature of LLMs themselves. These models operate as black boxes, and their internal reasoning processes are not readily inspectable. As the prompt sensitivity analysis showed, small differences in phrasing can produce measurable variation in output quality, yet the source of these differences remains difficult to diagnose or mitigate. This contrasts with traditional optimisation or statistical learning methods, where model parameters and logic are explicit. The opacity of LLM-based synthesis complicates efforts to guarantee consistency, reproducibility, or auditability, particularly when models are updated or substituted. Reproducibility depends on access to specific model versions, which may be deprecated or modified over time. To mitigate this, the exact model snapshots used in this study are documented in Table 4, and the codebase includes scripts that can be used to benchmark alternatives.

The framework also does not enforce higher-order joint constraints during generation. Although some multivariate relationships are recovered implicitly, this is not guaranteed. As documented in Section 5.3, relational artifacts may persist in the output, including a bimodal partner age gap and underrepresentation of same-age couples, likely reflecting the absence of targeted relational feedback in the prompt.

## 6.7 Future work

Future work will focus on extending the framework to produce spatially explicit and behaviourally rich synthetic populations suitable for integration into agent-based models. This includes assigning households to dwellings in a way that reflects local sociodemographic patterns, linking individuals to workplaces, and generating daily activity schedules informed by personal and household characteristics such as age, occupation and car availability. These additions are particularly important for domains such as evacuation modelling, where individual decisions are shaped not only by demographic attributes, but also by mobility resources and role-based obligations. Further development will also explore the

construction of social networks beyond the household, enabling more realistic modelling of interpersonal interactions and information diffusion.

While this study focused on a limited set of demographic variables, extending the framework to support a richer set of attributes remains an open area of investigation. Many population synthesis methods are known to struggle with high-dimensional attribute spaces due to sparsity and dependence between variables. Further work is needed to understand how LLMs will perform and whether they will maintain coherence or exhibit instability as the number of variables increases. Evaluating this behaviour systematically will be important for understanding the limits of the framework and for guiding its use in more complex modelling scenarios.

Two approaches are suggested that may help to mitigate the geographic and cultural biases introduced by the LLM's pre-training data. First, prompt augmentation could improve the model's demographic priors for underrepresented regions. Regional context could be supplied manually by the user or retrieved automatically from publicly available sources such as Demographic Health Surveys or UN statistics. Second, the synthetic population could serve as pseudo-microdata for a subsequent reweighting step using a method such as Iterative Proportional Updating, allowing classical fitting techniques to correct residual distributional errors after generation. The effectiveness of this hybrid approach would depend on the LLM having generated a sufficiently diverse pool of household structures for the reweighting to act upon.

Finally, the relational artifacts documented in Section 5.3, including a bimodal partner age gap and underrepresentation of same-age couples, suggest that targeted refinements to the prompt design could improve intra-households age structure. Incorporating soft priors for key age relationships, either as explicit instructions or as additional feedback signals within the generation loop, represent a natural next step.

## 7 Conclusion

This paper presents a sample-free framework for generating household-structured synthetic populations using LLMs. Model agnostic, the method requires only marginal distributions and does not rely on microdata, cross-tabulations or fine-tuning. By formulating population synthesis as a constrained reasoning task, it produces populations that are demographically plausible and internally coherent.

The framework was tested across 109 countries and two detailed case studies in Newcastle upon Tyne and Dar es Salaam, demonstrating strong generalisability across diverse demographic contexts. In the global evaluation, it achieved very close alignment on simpler marginals such as gender and household size, while more structurally complex attributes such as household composition and age were also reproduced with good accuracy. The framework provided viable populations even when input data were sparse or inconsistent. The ability to reconcile conflicting constraints and generate plausible household structures makes it a practical approach, even in settings where conventional techniques are difficult to apply. Where microdata are available, sample-based methods such as IPU offer a faster and cheaper alternative; however, the results demonstrate that the LLM framework can achieve comparable distributional fit on most variables, and offers greater flexibility in variable definitions. This has direct implications for a wide range of agent-based modelling applications, including applications in transport, health and disaster management. Population realism is essential for social simulations, where demographic characteristics influence how agents behave and respond to their environment. Attributes such as age, gender and marital status affect individual perceptions, preferences and risk tolerances. Household structures are equally important, as they define a person's responsibilities, resources and constraints. For example, in an evacuation scenario, decisions made by a single adult will differ from those of a caregiver responsible for young children or older relatives.

By enabling the creation of demographically structured populations from minimal inputs, this framework supports the wider application of agent-based models in contexts where detailed microdata are unavailable. It provides a flexible tool for generating realistic inputs for simulations used in planning, policy analysis and decision support.

## Supporting information

**S1 File. Supplementary materials.** Appendix A: example prompt, including distributional feedback data. Appendix B: preliminary experiment evaluating LLM querying without iterative distributional feedback. Appendix C: benchmarking results for all evaluated models, including age-gender population pyramids and distributional fit metrics. Appendix D: global case study results, including age-gender population pyramids and a comparison of SRMSE and Wasserstein metrics. Appendix E: scatter plots for univariate OLS regression analyses of country-level SRMSE predictors. Appendix F: meta-prompt used to generate prompt variants for sensitivity analysis.
(DOCX)

## Author contributions

**Conceptualization:** Michael Jones, Richard Dawson, Jon Mills.

**Data curation:** Michael Jones.

**Formal analysis:** Michael Jones.

**Investigation:** Michael Jones.

**Methodology:** Michael Jones.

**Software:** Michael Jones.

**Supervision:** Richard Dawson, Jon Mills.

**Validation:** Michael Jones.

**Visualization:** Michael Jones.

**Writing – original draft:** Michael Jones.

**Writing – review & editing:** Michael Jones, Richard Dawson, Jon Mills.

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
