## [Decision Letter · Decision Letter 0]

22 Feb 2026

PONE-D-26-01425A large language model framework for sample-free population synthesisPLOS One

Dear Dr. Authors,

Thank you for submitting your manuscript to PLOS ONE. After careful consideration, we feel that it has merit but does not fully meet PLOS ONE’s publication criteria as it currently stands. Therefore, we invite you to submit a revised version of the manuscript that addresses the points raised during the review process.

We look forward to receiving your revised manuscript.

Kind regards,

Mohammad Salah Hassan, Ph.D

Academic Editor

PLOS One

Journal Requirements:

4. We note that Figure 1 your submission contain [map/satellite] images which may be copyrighted. All PLOS content is published under the Creative Commons Attribution License (CC BY 4.0), which means that the manuscript, images, and Supporting Information files will be freely available online, and any third party is permitted to access, download, copy, distribute, and use these materials in any way, even commercially, with proper attribution. For these reasons, we cannot publish previously copyrighted maps or satellite images created using proprietary data, such as Google software (Google Maps, Street View, and Earth). For more information, see our copyright guidelines: http://journals.plos.org/plosone/s/licenses-and-copyright.

(1) You may seek permission from the original copyright holder of Figure 1 to publish the content specifically under the CC BY 4.0 license.

Additional Editor Comments:

Dear Dr. Authors,

Thank you for submitting your manuscript to PLOS ONE. Your paper has now been reviewed by three experts, and I appreciate your patience during the evaluation process.

All three reviewers found the work to be technically sound and clearly written. They recognized the originality of applying a large language model to sample-free population synthesis and agreed that the approach addresses an important methodological challenge, particularly for data-scarce settings. The global evaluation across 109 countries and the case studies in Newcastle and Dar es Salaam were viewed as significant strengths, and the reviewers appreciated the transparency with which you discussed prompt sensitivity and model behavior.

At the same time, one reviewer raised several substantive concerns related to methodological clarity, reproducibility, and reporting. These comments do not question the core contribution of the study, but they do highlight areas where the manuscript would benefit from greater transparency and additional explanation. In particular, clearer reporting of uncertainty across repeated runs, more explicit documentation of model versions and decoding parameters, and a fuller explanation of how conflicting input marginals are reconciled within the feedback loop would strengthen the paper considerably. Additional discussion around multivariate validation, benchmarking context, and reproducibility would also improve the robustness and long-term value of the work.

In light of these comments, the decision at this stage is Major Revision. The overall tone of the reviews is constructive and supportive, and I believe the requested revisions are achievable. With careful attention to the reviewers’ feedback, the manuscript has strong potential to make a meaningful contribution to the literature.

When you submit your revised manuscript, please include a detailed response explaining how you have addressed the reviewers’ comments.

Thank you again for the opportunity to consider your work. I look forward to receiving your revision.

Kind regards,

Reviewers' comments:

Reviewer's Responses to Questions

**Comments to the Author**

1. Is the manuscript technically sound, and do the data support the conclusions?

Reviewer #1: Yes

Reviewer #2: Yes

Reviewer #3: Yes

2. Has the statistical analysis been performed appropriately and rigorously? 

Reviewer #1: Yes

Reviewer #2: Yes

Reviewer #3: Yes

3. Have the authors made all data underlying the findings in their manuscript fully available?

Reviewer #1: Yes

Reviewer #2: Yes

Reviewer #3: Yes

4. Is the manuscript presented in an intelligible fashion and written in standard English?

Reviewer #1: Yes

Reviewer #2: Yes

Reviewer #3: Yes

5. Review Comments to the Author

Reviewer #1: Allow me start with an appreciation for the opportunity to review this great piece.

Your work is valuable and top notch however I have some few comments that I believe could even make it better and these are as follows from the different sections:

Review comments

Abstract:

Line 9 and 10: is it the traditional synthesis methods that heavily rely on micro data? If yes, then you could contextualize it other than leaving it open. What makes micro data unavailable? Is it restrictions due to privacy? May be you could add it’s not being available in many cases, the restriction due to privacy or being obsolete.

Line 15: Criteria for training the LLM is not visible. Which kind of data was used in the pre-training stage? Other than saying the model “draws on prior knowledge encoded in its training to propose plausible attribute combinations”, could you include how the training was done?

Lines 20 and 21: besides results from the different studies demonstrating applicability of the framework in data rich and scarce environments, what are the major conclusions and recommendations you draw from testing the model.

Background

Line 96: how about if “….. introduced an iterative process that builds households one member at a time” is rewritten as “….. introduced an iterative process that builds households member by member”.

Methods

Line 180: crosstabulation to cross tabulation or cross-tabulation.

Line 279 to 282: What are the 3 dimensions for evaluating outputs? If they are distributional fit to the target descriptors, structural feasibility of the generated records, and computational efficiency in terms of runtime and token cost then what is statistical accuracy, generation speed and affordability? Do they mean the same things? Could it be possible to be consistent and also add parameters in table 3 here to show which one belongs to which dimension.

Results

Lines 331 to 334: GPT-40 was selected as the best model overall but GRO-40 mini scored better in total cost, time and they both tied in success rate. For a limited resource setting one could opt for GRO-40 mini even with the lower accuracy. Could it be possible to further justify the selection of GRO-40 other than just saying it demonstrated the best balance between distributional fit, speed and reliability. If there is any kind of weight attached to the different parameters of performance measures may be it would be better.

Line 331, and 410: says “For Newcastle and Dar es Salaam, larger populations of approximately 100,000 individuals were generated to enable detailed evaluation of internal structure and feasibility”. Table 3 measures performance on a 500 household test case. Why use 500 households for model selection yet test cases had 100,000 individuals and convergence reached at about 800 households.

Just out of curiosity, do the results in table 3 remain the same if you use 800 households when convergence occurs to measure LLM performance?

Discussion

Line 447: double space between demographic and characteristics; reduce to single space.

Limitations

Line 538 to 543: it is great to note that the internal thinking process of LLM is not inspectable which complicates consistency, reproducibility and auditability.

In line 99 to 100 which says, “sample free models become increasingly complex as more attributes are added stated” could this be part of the limitations? If yes, add it to the limitations and how it was handled.

Conclusion

Line 566 to 568: how about results with SRME are kept in the results section and we have conclusions as typical conclusions.

Line 568 to 568: how about you revised “The framework was shown to provide viable populations even when input data were sparse or inconsistent” to “The framework provided viable populations even when input data were sparse or inconsistent.”. This uses active voice and makes it more direct.

Reviewer #2: The manuscript presents a model agnostic and sample free LLM framework for generating household structured synthetic populations using only aggregate demographic data. After benchmarking several models, the authors select GPT 4o and demonstrate strong performance on simple marginals across 109 countries and in two detailed case studies (Newcastle and Dar es Salaam). More complex attributes show moderate accuracy. The study addresses an important methodological gap, particularly for data scarce environments, and offers potential value for applied epidemiological and clinical research. The literature review on synthetic data generation is timely and contributes to an underrepresented but widely used area of research.

Statistical and Methodological Considerations

1. Optimization and Feedback Mechanism

The optimization setup in Eq. (1) frames the task as minimizing discrepancies such as SRMSE and JSD. However, its relationship to the iterative prompting procedure is indirect. Because the LLM receives only summarized discrepancies, there is no assurance of consistent improvement or convergence. An ablation comparing generation with and without discrepancy feedback would clarify the role of the feedback loop.

2. Fit Metrics

SRMSE is an appropriate primary metric, but additional measures such as category weighted SRMSE or the Wasserstein distance for ordered variables (for example age) would strengthen the evaluation by reducing the influence of small categories and capturing ordering information.

3. Dependence on LLM Pretraining Data

Although the method does not rely on sample data, the demographic relationships produced by the LLM reflect its pretraining data. This may introduce geographic or cultural biases, particularly in regions with limited representation in the training corpus or with atypical demographic structures. A brief discussion of potential future solutions would enhance the manuscript.

4. Handling of Inconsistent Input Marginals

When input marginals originate from inconsistent or heterogeneous sources, as in Dar es Salaam, the LLM reconciles these conflicts implicitly. Without explicit weighting or error modeling, these adjustments may appear arbitrary. A defined conflict resolution policy would provide greater transparency.

5. Lack of Higher Order Constraints

The framework matches marginal distributions but does not enforce higher order joint relationships. Although some joint patterns appear reasonable, the absence of multivariate constraints may lead to artifacts, including unrealistic partner age gaps or household structures.

6. Reproducibility

Reproducibility depends on access to specific model versions, prompts, decoding parameters, and API behavior, which may change over time. The study would benefit from a fully scripted and version controlled pipeline.

7. Validation of Multivariate Structure

The manuscript suggests that LLMs guided only by marginals and feasibility rules can recover realistic multivariate dependencies. The Newcastle cross tab results support this, but stronger evidence would require quantifying uncertainty and validating held out relationships. Bootstrapping across multiple runs and evaluating held out descriptors would demonstrate stability and generalization.

8. Intra Household Age Patterns

The intra household age relationships show general plausibility but also include artifacts such as a bimodal partner age gap and underrepresentation of same age couples. These issues appear related to prompt design and the absence of targeted feedback on relational attributes. Incorporating soft priors for key relationships may improve performance.

9. Model Benchmarking

The benchmarking in Table 3 is helpful but incomplete for a methodological paper. Reporting failure modes, decoding settings, prompt lengths, and tokenization effects would provide a more comprehensive comparison. Reproducible scripts for re benchmarking would support long term validity.

10. Transparency of Prompts

Appendix A provides one example, but the exact prompts used in all analyses (global, Newcastle, Dar es Salaam) should be disclosed. Archiving versioned prompts, if possible, would increase transparency.

11. Determinants of Global Performance

The global evaluation summarizes SRMSE distributions and maps performance across countries. A regression of SRMSE on data quality indicators (such as year of source, percentage unknown, and category counts) and demographic characteristics (such as dependency ratios or migration proxies) would support the claim that data quality and unusual demographic profiles drive errors.

12. Conflict Reconciliation in Dar es Salaam

The framework converges to intermediate values when marginals conflict, which is reasonable. However, this behavior indicates the need for an explicit reconciliation strategy, possibly involving weighting based on survey precision or recency.

13. Privacy and Re identification Risk

Although the approach is sample free, LLMs may reproduce memorized or rare patterns. The study would be strengthened by:

a. quantifying the uniqueness or distance of synthetic records relative to available microdata,

b. checking for potential verbatim leakage when prompts include specific geographic identifiers, and

c. including an ethics statement covering LLM safety and guardrails.

14. Comparison With Established Methods

A stronger methodological contribution would include direct comparisons with established synthetic population methods, such as GenSynthPop, IPF or IPU, and Bayesian networks, using identical metrics for Newcastle and Dar es Salaam.

15. Cost and Scalability Analysis

The manuscript notes linear cost scaling and provides a cost estimate for the UK. Including empirical measurements of throughput by batch size, experiments with multi household generation per call, and evaluation of cache aware prompting would clarify the trade offs between cost and performance.

Writing enhancements

16. The manuscript uses “sex” and “gender” interchangeably; adopting consistent terminology throughout, including in figures, tables, and prompts, would improve clarity.

17. The “other” household category is heterogeneous, providing a brief taxonomy or mapping would improve clarity for readers.

Reviewer #3: This paper proposes using an LLM to generate household-structured synthetic populations directly from aggregate demographic data, without requiring micro-data. The method runs an iterative prompt-and-feedback loop (Algorithm 1, page 10) that steers generation toward target marginals while enforcing household validity through schema and rule checks.

The evaluation covers 109 countries for breadth, Newcastle as a clean-data benchmark, and Dar es Salaam to stress-test the method under fragmented, partially contradictory inputs. Table 4 shows SRMSE from 0.003 (sex) to 0.157 (age), which is reasonable given complexity differences. The benchmarking of 10 LLMs in Table 3 is a practical addition. I appreciated the honesty about prompt sensitivity in Section 6.4 — many LLM papers skip this.

My main concern is the lack of variance reporting. The 20-run experiments (Sections 5.2.2, 5.3) show variation in Figures 5 and 10, but without standard deviations or confidence intervals it is hard to tell whether differences between models or settings are real or just noise.

Revision points:

1. Report SDs or 95% CIs for SRMSE across repeated runs, for both the global and Newcastle experiments.

2. State exact model snapshots (e.g. GPT-4o-2024-11-20), temperature, and top-p. Section 6.4 shows these matter.

3. Date the cost figures in Table 3 and add a cost-per-household metric.

4. Finalize the data repository with a DOI before acceptance. List what it contains (processed marginals, prompts, configs, eval scripts, example outputs).

5. In Dar es Salaam, household size says 14.5% one-person households while composition says 9.6%, and the model lands on 12.8%. Explain how the feedback loop balances competing targets. Are all marginals weighted equally? This will come up in any real-world application with inconsistent inputs.

6. Report the rate of structurally invalid households (e.g. child-aged spouse) before and after the validation step. This would show how much work the LLM does versus the rule checker.

7. Figure 4 (25 pyramids on one page) is hard to read in print. Move some to supplementary.

8. Figure 12 has missing bars that the text explains as generation failures. Add this to the caption.

9. References 44-45 are both World Population Prospects — combine into one.

10. Reference 42 (Lim et al.) is still on arXiv. Update if published.

11. Fix “occassion” (p. 23) and standardize hyphenation of “sample-based.”

A note for the discussion: the systematic age and composition biases could matter for downstream transport, health, or disaster simulations. A sentence connecting these to application risks would help practitioners.

No ethics concerns. Aggregate public data, no human participants.

6. PLOS authors have the option to publish the peer review history of their article (what does this mean?). If published, this will include your full peer review and any attached files.

Reviewer #1: **Yes:** Tom Egimu

Reviewer #2: **Yes:** Sreejata Dutta

Reviewer #3: No

---

## [Author Response · Author response to Decision Letter 1]

31 Mar 2026

Editor

2. All author-generated code to be made available without restrictions upon publication of the work.

Both repositories are publicly available without restriction. Links are provided in Section 1 of the manuscript: the population generation library at https://github.com/MJones235/LLM-Population-Generator/releases/tag/v1.0.0 and the data collection and processing scripts at https://github.com/MJones235/Synthetic-Population-Experiments/releases/tag/v1.0.0.

3. We strongly recommend all authors decide on a data sharing plan before acceptance.

A data repository has been established at data.ncl.ac.uk and assigned a DOI: https://doi.org/10.25405/data.ncl.31830205. The repository contains processed marginals, all prompts used across the global, Newcastle, and Dar es Salaam analyses, configuration files, and example outputs. The repository is currently under review for publication, so the DOI is not yet active. In the interim, it can be viewed using the following private link: https://figshare.com/s/9a0e2585833c9f75e0ad.

4. We note that Figure 1 your submission contain [map/satellite] images which may be copyrighted.

Figure 1 does not use satellite imagery. All maps are generated programmatically using Plotly. Country boundaries are obtained from Natural Earth, which releases all data into the public domain.

5. If the reviewer comments include a recommendation to cite specific previously published works, please review and evaluate these publications to determine whether they are relevant and should be cited

Not applicable

Reviewer 1

Abstract:

Line 9 and 10: is it the traditional synthesis methods that heavily rely on micro data? If yes, then you could contextualize it other than leaving it open. What makes micro data unavailable? Is it restrictions due to privacy? May be you could add it’s not being available in many cases, the restriction due to privacy or being obsolete.

The abstract has been revised to clarify that it is the established synthesis methods that are dependent on microdata, and to contextualise why microdata are often unavailable: they are infrequently collected on a decadal cycle, restricted for privacy protection, and typically released only as small public-use samples at coarse geographic scales.

Line 15: Criteria for training the LLM is not visible. Which kind of data was used in the pre-training stage? Other than saying the model “draws on prior knowledge encoded in its training to propose plausible attribute combinations”, could you include how the training was done?

The abstract has been updated to use "pre-training" rather than "training" to avoid implying that the LLM was fine-tuned for this task. Section 3.6 has been revised to clarify that the framework uses pre-trained foundation models without modification. This is a deliberate design choice to minimise data requirements and ensure the framework is accessible to practitioners.

Lines 20 and 21: besides results from the different studies demonstrating applicability of the framework in data rich and scarce environments, what are the major conclusions and recommendations you draw from testing the model.

The abstract has been revised to state explicitly that the principal contribution of the framework is to enable population synthesis in data-constrained settings, expanding the applicability of agent-based modelling.

Background

Line 96: how about if “….. introduced an iterative process that builds households one member at a time” is rewritten as “….. introduced an iterative process that builds households member by member”.

Revised as suggested.

Methods

Line 180: crosstabulation to cross tabulation or cross-tabulation.

Revised to "cross-tabulation" throughout the manuscript.

Line 279 to 282: What are the 3 dimensions for evaluating outputs? If they are distributional fit to the target descriptors, structural feasibility of the generated records, and computational efficiency in terms of runtime and token cost then what is statistical accuracy, generation speed and affordability? Do they mean the same things? Could it be possible to be consistent and also add parameters in table 3 here to show which one belongs to which dimension.

The evaluation framework has been revised to use three consistent dimensions throughout: (i) distributional fit, (ii) structural feasibility, and (iii) computational efficiency. The paragraph introducing Table 4 has been updated to map each column explicitly to its corresponding dimension.

Results

Lines 331 to 334: GPT-40 was selected as the best model overall but GRO-40 mini scored better in total cost, time and they both tied in success rate. For a limited resource setting one could opt for GRO-40 mini even with the lower accuracy. Could it be possible to further justify the selection of GRO-40 other than just saying it demonstrated the best balance between distributional fit, speed and reliability. If there is any kind of weight attached to the different parameters of performance measures may be it would be better.

The selection of GPT-4o over GPT-4o-mini is further justified in Appendix C, which shows the age distribution produced by each model. The difference in JSD translates to a visually noticeable degradation in distributional quality for GPT-4o-mini. While GPT-4o-mini achieved lower cost, its distributional fit was inferior to GPT-4o, making it unsuitable where accuracy is the primary criterion.

Line 331, and 410: says “For Newcastle and Dar es Salaam, larger populations of approximately 100,000 individuals were generated to enable detailed evaluation of internal structure and feasibility”. Table 3 measures performance on a 500 household test case. Why use 500 households for model selection yet test cases had 100,000 individuals and convergence reached at about 800 households.

Just out of curiosity, do the results in table 3 remain the same if you use 800 households when convergence occurs to measure LLM performance?

The benchmarking experiment was repeated using 800 households, consistent with the convergence point identified in Section 5.3. Relative model performance was largely unchanged. Section 3.6 and Table 4 have been updated accordingly.

Discussion

Line 447: double space between demographic and characteristics; reduce to single space.

Done

Limitations

Line 538 to 543: it is great to note that the internal thinking process of LLM is not inspectable which complicates consistency, reproducibility and auditability.

Thank you - no change required

In line 99 to 100 which says, “sample free models become increasingly complex as more attributes are added stated” could this be part of the limitations? If yes, add it to the limitations and how it was handled.

This point is addressed in Section 6.7 as a direction for future work rather than a confirmed limitation. The study focuses on a limited set of demographic variables, and while other methods are known to struggle with high-dimensional attribute spaces, this has not yet been established as a limitation of the LLM-based approach. Further investigation is left as a suggestion for future work.

Conclusion

Line 566 to 568: how about results with SRME are kept in the results section and we have conclusions as typical conclusions.

The conclusion has been revised to avoid reporting specific numerical results, instead summarising findings in qualitative terms.

Line 568 to 568: how about you revised “The framework was shown to provide viable populations even when input data were sparse or inconsistent” to “The framework provided viable populations even when input data were sparse or inconsistent.”. This uses active voice and makes it more direct.

Revised as suggested.

Reviewer 2

Statistical and Methodological Considerations

1. Optimization and Feedback Mechanism

The optimization setup in Eq. (1) frames the task as minimizing discrepancies such as SRMSE and JSD. However, its relationship to the iterative prompting procedure is indirect. Because the LLM receives only summarized discrepancies, there is no assurance of consistent improvement or convergence. An ablation comparing generation with and without discrepancy feedback would clarify the role of the feedback loop.

A preliminary experiment was conducted in which households were generated without distributional feedback — the prompt contained the target distributions but no information about the population generated so far. As shown in Figure D1 (Appendix D), distributional fit was poor under this configuration: household composition was dominated by "couple with dependent children" households (>60% of generated records against a 17% target), household size significantly overrepresented 3–4 persons, and the age distribution collapsed to working-age adults and children. These results confirm that the iterative feedback mechanism is essential to achieving distributional alignment, and that supplying target distributions within the prompt alone is insufficient. The justification for this design choice has been placed in Section 3.4, where the feedback algorithm is described, and full details of the preliminary experiment are provided in Appendix D.

2. Fit Metrics

SRMSE is an appropriate primary metric, but additional measures such as category weighted SRMSE or the Wasserstein distance for ordered variables (for example age) would strengthen the evaluation by reducing the influence of small categories and capturing ordering information.

The relationship between SRMSE and Wasserstein distance for age across 109 countries is discussed in Section 5.2.1 and illustrated in Figure E1, Appendix E. The two metrics diverge for a subset of countries, with the direction of divergence related to the shape of the population pyramid. Countries with expansive pyramids, such as Nigeria, Madagascar and Sudan, tend to have high SRMSE relative to their Wasserstein distance, as category-level errors in these distributions are penalised more heavily by SRMSE. The opposite pattern is observed for countries with constrictive pyramids, such as Italy, Portugal and Canada. The implications of this metric sensitivity are discussed in Section 6.2: Africa appears to underperform relative to other continents under SRMSE, but this difference is considerably reduced when Wasserstein distance is used instead. Which metric better reflects fitness for purpose will depend on the intended application.

3. Dependence on LLM Pretraining Data

Although the method does not rely on sample data, the demographic relationships produced by the LLM reflect its pretraining data. This may introduce geographic or cultural biases, particularly in regions with limited representation in the training corpus or with atypical demographic structures. A brief discussion of potential future solutions would enhance the manuscript.

The potential for geographic and cultural bias introduced by the LLM's pre-training data is discussed in Section 6.2, where performance is evaluated across demographic regimes and regions. Two mitigation strategies are proposed in Section 6.7. First, prompt augmentation could improve the model's demographic priors for underrepresented regions, with regional context supplied either manually or retrieved automatically from publicly available sources such as Demographic Health Surveys or UN statistics. Second, the synthetic population could serve as pseudo-microdata for a subsequent IPU reweighting step, allowing classical fitting techniques to correct residual distributional errors after generation. The effectiveness of this hybrid approach would depend on the LLM having generated a sufficiently diverse pool of household structures for the reweighting to act upon.

4. Handling of Inconsistent Input Marginals

When input marginals originate from inconsistent or heterogeneous sources, as in Dar es Salaam, the LLM reconciles these conflicts implicitly. Without explicit weighting or error modeling, these adjustments may appear arbitrary. A defined conflict resolution policy would provide greater transparency.

Two prompt configurations were tested for Dar es Salaam: a baseline (E1) presenting all marginals without prioritisation or conflict resolution guidance, and a modified version (E2) in which explicit instructions were added to prioritise household size alignment above other marginals. Contrary to expectation, mean household size was identical under both configurations (3.87 persons in each case), suggesting that the added instructions did not alter the model's sampling behaviour for this variable. Fit to the remaining distributions deteriorated under E2, with mean SRMSE increasing from 0.107 to 0.159.

These results suggest that conflict resolution is better handled as a preprocessing step prior to generation, where targets can be explicitly reconciled before being passed to the model, rather than through in-prompt instructions that require the LLM to simultaneously interpret inconsistent targets and generate plausible households. This analysis (and consequently the text for Prompt 2) was not included in the main manuscript as the experiment was inconclusive with respect to the original question, and a more systematic prompt comparison would be needed to draw firm conclusions. The baseline configuration E1 was retained for all subsequent analyses and is reproduced in Appendix A.

Prompt E2 is reproduced below:

INSTRUCTIONS:

1. Review the current and target distributions for household size, age, and gender (provided below).

2. Select a household size.

- Alignment to the target mean is the top priority

- Prefer under-represented sizes

3. Choose household composition after size is fixed. Fitting this distribution is lower priority.

- 1 person: contains a single adult of any age.

- 2 person: can be either a couple, a lone parent with a child, or unrelated housemates.

- 3+ person: can be a family (couples or lone parents with children, extended family), or unrelated housemates.

4. Select individuals for the household, prioritizing under-represented age groups and genders.

5. For each person, include:

- age (0–120)

- gender ("Male" or "Female")

- relationship (first must be "Head" and be an adult; valid values: "Spouse", "Partner", "Child", "Parent", "Sibling", "Grandchild", "Grandparent", "Housemate", "Lodger", "Aunt", "Uncle", "Nephew", "Niece", "Cousin", "Child-in-law", "Parent-in-law", "Sibling-in-law")

6. Output only a JSON object with a "household" array containing these individuals. Do not include markdown, formatting, or explanations.

5. Lack of Higher Order Constraints

The framework matches marginal distributions but does not enforce higher order joint relationships. Although some joint patterns appear reasonable, the absence of multivariate constraints may lead to artifacts, including unrealistic partner age gaps or household structures.

A dedicated paragraph has been added to Section 6.6 acknowledging that the framework does not enforce higher-order joint constraints during generation. Although some multivariate relationships are recovered implicitly, this is not guaranteed. Specific relational artifacts documented in Section 5.3, including a bimodal partner age gap and underrepresentation of same-age couples, are noted as likely reflecting the absence of targeted relational feedback in the prompt.

6. Reproducibility

Reproducibility depends on access to specific model versions, prompts, decoding parameters, and API behavior, which may change over time. The study would benefit from a fully scripted and version controlled pipeline.

All code is available in version-controlled public repositories, with links provided in the introduction of the manuscript. The exact model snapshots used in each experiment are documented in Table 4, and the codebase includes scripts for testing the framework with alternative model versions. A note on reproducibility has been added to Section 6.6 acknowledging that access to specific model versions cannot be guaranteed as APIs evolve, and directing readers to the documented workflow for substituting alternative versions.

7. Validation of Multivariate Structure

The manus

---

## [Decision Letter · Decision Letter 1]

20 Apr 2026

PONE-D-26-01425R1A large language model framework for sample-free population synthesisPLOS One

Dear Dr. Jones,

Thank you for submitting your manuscript to PLOS ONE. After careful consideration, we feel that it has merit but does not fully meet PLOS ONE’s publication criteria as it currently stands. Therefore, we invite you to submit a revised version of the manuscript that addresses the points raised during the review process.

We look forward to receiving your revised manuscript.

Kind regards,

Mohammad Salah Hassan, Ph.D

Academic Editor

PLOS One

Journal Requirements:

Additional Editor Comments:

Dear Authors,

Thank you for submitting your revised manuscript. The revision is clearly much stronger, and the additional analyses and clarifications have addressed the main substantive concerns raised during review. The methodological contribution is now much clearer, and I appreciate the effort that has gone into the revision.

Before the manuscript can be finalized, however, I would ask you to address a small number of remaining issues in the file and submission materials.

First, please resolve the data availability wording so that it is fully clear and internally consistent. At present, the submission states that all data and code are publicly available, while also noting that the repository DOI is not yet active and that access is currently via a private link. Please make sure the repository status, DOI, and access conditions are presented in a way that is fully consistent with the journal’s data availability requirements.

Second, the manuscript and accompanying statements would benefit from one final careful proofreading pass. At least one typo remains in the ethics statement, and there are still a few minor language issues that should be corrected before the paper moves forward. Please review the manuscript closely for spelling, grammar, and phrasing.

Third, please ensure that all journal style and technical requirements have been fully satisfied, including any file-formatting and submission-format requirements noted by the editorial office.

These are minor points and do not require another substantive revision of the work itself, but they should be addressed carefully so that the final version is clean and publication-ready.

Best regards,

Reviewers' comments:

Reviewer's Responses to Questions

**Comments to the Author**

1. If the authors have adequately addressed your comments raised in a previous round of review and you feel that this manuscript is now acceptable for publication, you may indicate that here to bypass the “Comments to the Author” section, enter your conflict of interest statement in the “Confidential to Editor” section, and submit your "Accept" recommendation.

Reviewer #1: All comments have been addressed

Reviewer #2: All comments have been addressed

2. Is the manuscript technically sound, and do the data support the conclusions?

Reviewer #1: Yes

Reviewer #2: Yes

3. Has the statistical analysis been performed appropriately and rigorously? 

Reviewer #1: Yes

Reviewer #2: Yes

4. Have the authors made all data underlying the findings in their manuscript fully available?

Reviewer #1: Yes

Reviewer #2: Yes

5. Is the manuscript presented in an intelligible fashion and written in standard English?

Reviewer #1: Yes

Reviewer #2: Yes

6. Review Comments to the Author

Reviewer #1: It has been great having the opportunity to review your work. You have even made it better and we in the limited resource setting who suffer so much from missingness of micro data will greatly benefit from it.

Reviewer #2: All of my major statistical and methodological concerns have been fully addressed. The authors conducted additional experiments, added uncertainty quantification, strengthened benchmarking, and were appropriately cautious in interpreting limitations. The revised manuscript is substantially stronger, methodologically rigorous, and suitable for publication.

7. PLOS authors have the option to publish the peer review history of their article (what does this mean?). If published, this will include your full peer review and any attached files.

Reviewer #1: No

Reviewer #2: **Yes:** Sreejata Dutta

---

## [Author Response · Author response to Decision Letter 2]

29 Apr 2026

Typographical corrections

A number of typographical errors identified during final proofreading have been corrected:

• "Tables 4" has been corrected to "Table 4" (Section 6.1);

• "along" has been corrected to "alone" in the description of domain-specific validation rules (Section 6.1);

• "addition" has been corrected to "additional" in the caption to Table 2;

• "inference were" has been corrected to "inference times were" (Section 6.1);

• "that reflects" has been corrected to "that reflect" in the list of contributions (Section 1);

• "established" has been corrected to "establishes" in the methodology overview (Section 3.1);

• the confidence interval "[0.465, 0661]" has been corrected to "[0.465, 0.661]" in Table 8;

• a missing comma has been inserted after citation [20,21] in Section 2;

• the cross-reference "Figure 12" in Section 6.4 has been corrected to "Figure 13";

• the double full stop in the caption to Figure E4 has been removed; and,

• "appesar" has been corrected to "appear" in the ethics statement.

Reference update

Reference [42] (Lim et al.) has been updated to reflect its accepted publication in Transportation Research Part C: Emerging Technologies (2026, vol. 185, 105581; doi:10.1016/j.trc.2026.105581). The reference was previously cited as a preprint; the published version details have now been substituted.

Data availability statement

The data availability statement has been updated as the DOI is now live.

The statement reads: "All data and code supporting this study are publicly available. Processed marginals, prompts, configuration files, and example outputs are archived in the Newcastle University data repository (https://doi.org/10.25405/data.ncl.31830205). The population generation library is available at https://github.com/MJones235/LLM-Population-Generator/releases/tag/v1.0.0 and data collection and processing scripts at https://github.com/MJones235/Synthetic-Population-Experiments/releases/tag/v1.0.0."

---

## [Editor Report · Decision Letter 2]

4 May 2026

A large language model framework for sample-free population synthesis

PONE-D-26-01425R2

Dear Dr. Authors,

We’re pleased to inform you that your manuscript has been judged scientifically suitable for publication and will be formally accepted for publication once it meets all outstanding technical requirements.

Kind regards,

Mohammad Salah Hassan, Ph.D

Academic Editor

PLOS One

Additional Editor Comments (optional):

Dear Authors,

Thank you for submitting the revised manuscript. The remaining corrections appear to have been addressed, including the typographical edits, reference update, and revised data availability statement.

At this stage, no further scientific revisions are required. The manuscript may proceed subject to the journal’s final editorial and production checks.

Kind regards,
---

## [Editor Report · Acceptance letter]

PONE-D-26-01425R2

PLOS One

Dear Dr. Jones,

I'm pleased to inform you that your manuscript has been deemed suitable for publication in PLOS One. Congratulations! Your manuscript is now being handed over to our production team.

Kind regards,

on behalf of

Dr. Mohammad Salah Hassan

Academic Editor

PLOS One